# Optimizing Test-Time Compute via Meta Reinforcement Finetuning

Yuxiao Qu [* 1]   Matthew Y. R. Yang [* 1]   Amrith Setlur [1]   Lewis Tunstall [2]   Edward Emanuel Beeching [2]
Ruslan Salakhutdinov [1]   Aviral Kumar [1]

## Abstract

Training models to effectively use test-time compute is crucial for improving the reasoning performance of LLMs. Current methods mostly do so via fine-tuning on search traces or running RL with 0/1 outcome reward, but do these approaches efficiently utilize test-time compute? Would these approaches continue to scale as the budget improves? To answer these questions, in this paper, we formalize the problem of optimizing test-time compute as a meta-reinforcement learning (RL) problem, which provides a principled perspective on spending test-time compute. This perspective enables us to view the long output stream from the LLM as consisting of several episodes run at test time and leads us to use a notion akin to *cumulative regret* over output tokens as a way to measure the efficacy of test-time compute. Akin to how RL algorithms can best tradeoff exploration and exploitation over training, minimizing regret should also provide the best balance between exploration and exploitation in the token stream. While we show that state-of-the-art models do not minimize regret, one can do so by maximizing a *dense* reward bonus in conjunction with the outcome 0/1 reward RL. This bonus is the "progress" made by each subsequent block in the output stream, quantified by the change in the likelihood of eventual success. Using these insights, we develop **M**eta **R**einforcment Fine-**T**uning, or *MRT*, a new class of fine-tuning methods for optimizing test-time compute. *MRT* leads to a 2-3x relative gain in performance and roughly a 1.5x gain in token efficiency for math reasoning.

## 1. Introduction

Recent results in LLM reasoning (Snell et al., 2024) illustrate the potential to improve reasoning capabilities by scaling test-time compute. Generally, these approaches train models to produce traces that are longer than the typical correct solution, and consist of tokens that attempt to implement some "algorithm", *e.g.*, reflecting on previous answers (Qu et al., 2024; Kumar et al., 2024), planning (DeepSeek-AI et al., 2025), or implementing some form of linearized search (Gandhi et al., 2024). These approaches include explicitly fine-tuning pre-trained LLMs for algorithmic behavior, *e.g.*, SFT on search data (Gandhi et al., 2024; Nie et al., 2024), or running outcome-reward RL (DeepSeek-AI et al., 2025) against a 0/1 correctness reward.

While training models to spend test-time compute by generating long reasoning chains via outcome-reward RL has been promising, for continued gains from scaling test-time compute, we ultimately need to answer some critical understanding and method design questions. First, ***do current LLMs efficiently use test-time compute?*** That is, do they spend tokens roughly in the ballpark of the typical solution length or do they use too many tokens even on easy questions? Second, ***would LLMs be able to "discover" solutions to harder questions when run at much larger test-time token budgets than what was used for training***? Ultimately, we would want models to derive enough utility from every token (or any semantically meaningful segment) they produce, not only for efficiency but also because doing so imbues a systematic procedure to discover solutions for harder, out-of-distribution problems.

In this paper, we formalize the above challenges in optimizing test-time compute through the lens of ***meta reinforcement learning*** (RL) (Weng, 2019). To build our approach, we segment the output stream from an LLM on a problem into multiple *episodes* (Figure 2). If we were to only care about **(a) the efficiency**, then the LLM only needs to learn to *exploit* and directly output the answer without spending too many episodes. On the other hand, if the LLM is solely focused on **(b) the discovery**, then *exploration* is more desirable, so that the LLM can spend several episodes trying different approaches, verifying and revising them, before producing the final answer. Fundamentally, this is different from traditional RL since the goal here is to learn an LLM that implements explore-exploit algorithms on every test problem. In other words, we aim to learn such algorithms from training data, making this a ***meta*** RL learning problem.

---

[*]Equal contribution  [1]CMU  [2]Hugging Face. Correspondence to: Yuxiao Qu <yuxiaoq@andrew.cmu.edu>.

*Proceedings of the $42^{nd}$ International Conference on Machine Learning*, Vancouver, Canada. PMLR 267, 2025. Copyright 2025 by the author(s).

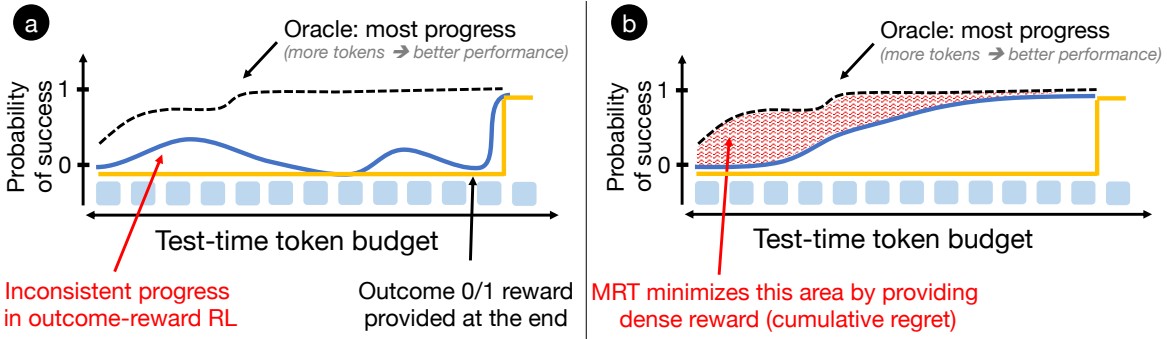

*Figure 1.* ***Standard outcome-reward reinforcement fine-tuning vs. MRT.*** Standard techniques for fine-tuning LLMs to use test-time compute optimize outcome reward at the end of a long trace. This does not incentivize the model to make use of intermediate tokens to make progress (*i.e.*, probability of eventual success) and leads to **1)** unnecessarily long output traces and **2)** inability to make steady progress on new, hard problems as shown in **(a)**. *MRT*, shown in **(b)**, trains the LLM to minimize cumulative regret over the entire output stream (red, shaded area) by optimizing a dense reward function in addition to sparse 0/1 reward and thus alleviates both challenges in **(a)**.

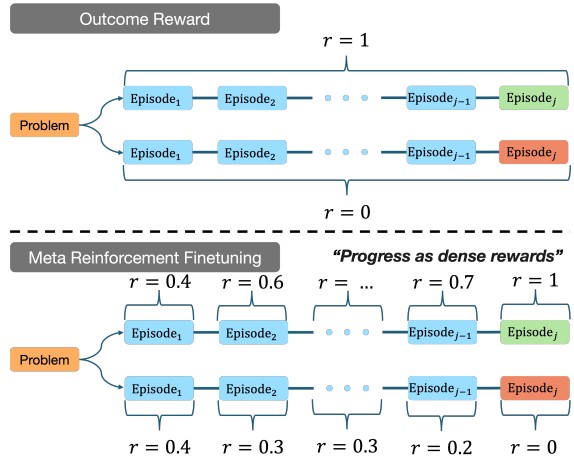

*Figure 2.* **MRT** uses dense rewards based on progress throughout the thinking trace (segmented into "episodes") to improve test-time efficiency and performance. Standard fine-tuning only trains models with outcome rewards at the end, thus reinforcing several traces that make subpar progress but somehow succeed (Figure 1(a)).

A desired "meta" behavior is one that strikes a balance between committing to an approach prematurely (*i.e.*, an "exploitation" episode) and trying too many high-risk strategies (*i.e.*, an "exploration" episode). From meta RL literature, we know that optimally trading off exploration and exploitation is equivalent to minimizing ***cumulative regret*** over the output token budget. This regret measures the difference between the likelihoods of success of the LLM and an oracle comparator, as illustrated by the red area in Figure 1(b).

By training an LLM to minimize cumulative regret on every query, we learn a reasoning strategy that is ***agnostic of the test-time budget***, *i.e.*, when deployed, the LLM spends only the necessary amount of tokens while still making progress when run at larger token budgets. We develop a new class of fine-tuning methods for optimizing test-time compute, which we refer to as ***Meta Reinforcement fine-Tuning (MRT)***, by minimizing the cumulative regret. Cumulative regret also provides a metric for evaluating the

effectiveness of SOTA reasoning models such as Deepseek-R1 (DeepSeek-AI et al., 2025) in using test-time compute.

In particular, we show that SoTA LLMs fine-tuned with outcome reward fail to improve their chances of discovering the right answer with more episodes, i.e., they do not make steady "progress" (illustration in Figure 1(a)), even though this behavior is critical for solving hard unseen problems. In fact, a much more naïve approach of running substantially fewer episodes coupled with majority voting is often more effective on harder questions in a FLOPs-matched evaluation (Figure 3). In contrast, we show that optimizing for progress in addition to outcome reward naturally emerges when the objective is to minimize regret. Concretely, our fine-tuning paradigm, ***MRT***, prescribes a dense reward bonus for RL training (Definition 4.1). This progress reward measures the change in the likelihood of finishing at a correct answer, before and after an episode is generated. Intuitively, the progress made by an episode is akin to the "information gained" about the underlying problem.

Empirically, we evaluate ***MRT*** in two settings that differ in the way they parameterize episodes. For the first setting, we employ the format of enclosing the reasoning process in between <think> markers and fine-tune base models: DeepScaleR-1.5B-Preview (Luo et al., 2025), DeepSeek-R1-Distill-Qwen-1.5B, and DeepSeek-R1-Distill-Qwen-7B (DeepSeek-AI et al., 2025), on a dataset of math reasoning problems. We find that ***MRT*** consistently outperforms outcome-reward RL, achieving state-of-the-art results at the 1.5B parameter scale across multiple benchmarks in aggregate (AIME 2024/2025, AMC 2023, *etc.*), with improvements in accuracy approximately of **2-3x** compared to those obtained by standard outcome-reward RL (GRPO), and 1.5–5x token efficiency over GRPO and base models. In the second setting, we fine-tune Llama3.1 models (3B and 8B) to implement backtracking search on math problems, where ***MRT*** achieves token efficiency improvements of **30%** over STaR (Zelikman et al., 2022) and **38%** GRPO.

We analyze *MRT* and show that it attains a lower cumulative regret and makes more steady progress, even when extrapolating to **2x** larger token budgets than what it was trained on. We also show that, unlike other methods for constraining length, which typically come at the cost of accuracy, *MRT* reduces the output length while boosting accuracy. We also find that the output length oscillates during RL and that length alone does not imply accuracy. Finally, we show that recipes for iteratively scaling test-time budgets–which have been noted to be more effective than training with a large output budget from scratch–also implicitly maximize progress and, hence, minimize regret.

## 2. Problem Formulation

In this section, we will formalize the problem of optimizing test-time compute as a meta RL problem. In the next section, we will show that this meta RL perspective can be used to evaluate if state-of-the-art models (*e.g.*, Deepseek-R1 (DeepSeek-AI et al., 2025)) are effectively and efficiently using test-time compute. Finally, we will utilize these ideas to develop a fine-tuning paradigm, called *MRT*, to optimize test-time compute.

### 2.1. Optimizing Test-Time Compute

We want an LLM to attain maximum performance on $\mathcal{P}_{\text{test}}$ within test-time budget $C_0$ (*i.e.*, $\forall \mathbf{x}$, $\mathbb{E}_{\mathbf{z} \sim \pi(\cdot|\mathbf{x})}|\mathbf{z}| \leq C_0$):

$$\max_{\pi} \ \mathbb{E}_{\mathbf{x} \sim \mathcal{P}_{\text{test}}, \mathbf{z} \sim \pi(\cdot|\mathbf{x})} \left[ r(\mathbf{x}, \mathbf{z}) \mid \mathcal{D}_{\text{train}} \right] \qquad (1)$$

While this is identical to optimizing the test performance like any standard ML algorithm, we emphasize that the budget $C_0$ used for evaluation is larger than the typical length of a correct response. This means that the LLM $\pi(\cdot|\mathbf{x})$ can afford to spend a part of the budget into performing operations that do not actually solve $\mathbf{x}$ but rather *indirectly* help the model in discovering the correct answer eventually. For example, consider a math proof question where the output is composed of a sequence of steps. If the policy could figure out that it should backtrack a few steps and restart its attempt, it may not only increase its chances of success, but also allow the LLM to confidently identify what steps to avoid and be careful about. However, compute budget $C_0$ does not necessarily equal to the deployment budget.

The conventional way of training an LLM to attain high outcome reward (DeepSeek-AI et al., 2025; Kimi-Team, 2025) given a fixed token budget is suboptimal. On problems where the typical solution length is well below the maximal token budget in training, this kind of training procedure would encourage redundancy and inefficient use of tokens as the model lacks incentive to develop more succinct responses. Now if the LLM is deployed with a budget less than the one used for training, yet sufficient to solve the task, the trained LLM might still not be able to finish responding.

While one way to address this issue is to force the model to terminate early if it can, this strategy is suboptimal for complex problems that require the model to potentially spend more budget on attempting to discover the right approach. In other words, training to succeed in the fewest tokens can spuriously cause the model to prematurely "commit" to an answer upon deployment, though this is not the best strategy. Additionally, training with only outcome reward is again suboptimal since it is unable to differentiate between solutions that are still on track progress and solutions that are not on track, if they both succeed or both do not succeed. We would instead like the model to still be rewarded positively for attempting to explore multiple approaches towards a solution and spending more tokens if it is on track and can succeed eventually. We therefore propose a different formulation for optimizing test-time compute that trains LLMs to be "optimal" at spending test-time compute, agnostic of the training token *budget* utilized, thus alleviating any commitment to a particular budget at test time.

***Budget-agnostic LLMs.*** The only approach that can guarantee optimal for any test-time compute budget is a *"budget-agnostic"* strategy that imbues behavior that can work well for multiple large enough budgets. To attain a high test performance, an LLM $\pi$ should exhibit behavior that trades off between exploration and exploitation to make the most use of the compute budget available.

### 2.2. Characterizing Optimal Use of Test-Time Compute

To develop a training paradigm to effectively use test-time compute, we first need to understand the characteristics of *budget-agnostic* LLMs that use test-time compute the most optimally. One way to characterize these LLMs is by explicitly segmenting the output stream $\mathbf{z} \sim \pi(\cdot|\mathbf{x})$ into a sequence of meaningful blocks (i.e., *episodes*), and viewing this sequence of episodes as some sort of an "adaptation" procedure on the test problem. This segmentation then allows us frame it as a meta-RL problem.

Formally, suppose that $\mathbf{z}$ can be divided into $k$ contiguous segments $\mathbf{z} \overset{\text{def}}{=} [\mathbf{z}_0, \mathbf{z}_1, \cdots, \mathbf{z}_{k-1}]$[1]. As shown in Figure 2, these episodes could consist of multiple attempts at a problem (Qu et al., 2024), alternating between verification and generation (Zhang et al., 2024) such that successive generation episodes attain better performance, or be paths in a search tree separated by backtrack markers.

We eventually want the LLM $\pi$ to succeed in the last episode it produces within the total budget, i.e., $\mathbf{z}_{k-1}$. However, since we operate in a setting where the LLM is unaware of the test-time deployment budget, we need to make sure that

---

[1]While there are many different strategies to segment $\mathbf{z}$ into variable number of episodes, for simplicity we assume a fixed number of episodes $k$ in our exposition. Note that if a particular $\mathbf{z}$ contains $l \geq k$ natural episodes, we can always choose to merge the last $l - k$ episodes into one for the purposes of our discussion.

the LLM is constantly making *progress* and is able to effectively strike the balance between *"exploration"*: producing tokens that are irrelevant to the final answer (*e.g.*, verifying previous steps or trying a different strategy), but *might* help in later episodes, and *"exploitation"*: attempting to simply expand on a approach to get to an answer.

Building on this intuition, our key insight is that the adaptation procedure implemented in the test-time token stream can be viewed as running an RL algorithm on the test problem, where prior episodes serve the role of "training" data for this purely in-context process. Under this abstraction, an "optimal" algorithm is one that makes steady progress towards discovering the solution for the problem with each episode, balancing between discovery and exploitation. As a result, we can use the metric of **cumulative regret** from RL to also quantify the optimality of this process.

---

**Definition 2.1** (*Cumulative regret*). Given $k$ episodes **z** generated from $\pi(\cdot|\mathbf{x})$, another LLM $\mu$ that computes an estimate of the correct response given episodes so far, and the optimal comparator policy given a $j$-episode budget as $\pi_j^*$, we define cumulative regret $\Delta_k^\mu(\mathbf{x}; \pi)$ as:

$$\mathbb{E}_{\mathbf{z}\sim\pi(\cdot|\mathbf{x})}\left[\sum_{j=0}^{k-1} J_r(\mathbf{x}; \pi_j^*) - J_r(\mathbf{x}; \mu(\cdot|\mathbf{x}, \mathbf{z}_{0:j}))\right].$$

---

Here $J_r$ is the expected 0/1 outcome reward attained by LLM $\mu$ when conditioning on prior episodes $\mathbf{z}_{0:j-1}$ produced by $\pi$, and $J_r(\pi^*)$ is the reward of the best possible budget-agnostic comparator $\pi^*$ that achievable via finetuning within a $j$-episode test-time budget. The ***meta-prover policy*** $\mu$ may be the same as or different from $\pi$. For example, if each episode produced by $\pi$ ends in an estimate of the answer, then we can measure 0/1 correctness of this answer in itself for computing $\Delta_k^\mu$ and set $\mu = \pi$. If some episodes produced by $\pi$ do not end in a final answer (e.g., episodes within the "think" block), we can use a different $\mu$ to help us extrapolate the answer. In our experiments, $\mu^2$ is the policy induced by the same underlying LLM, obtained by terminating the "think" block and forcing the model to estimate the best possible answer. ***The red colored area in Figure 1 denotes the cumulative regret.*** If the regret is large or if it increases with the number of episodes $k$, then we say that episodes **z** did not actually make meaningful progress. On the other hand, the lower the rate of growth in the regret, the more meaningful progress a budget-agnostic LLM $\pi$ makes as the budget grows.

## 3. Case Study: Analyzing DeepSeek-R1

Having defined the notion of cumulative regret, can we now use it to analyze state-of-the-art models, such as derivatives

---
[2]While $\mu$ and $\pi$ share the same underlying LLM, they represent distinct policies with different trajectory distributions.

of the DeepSeek-R1 (DeepSeek-AI et al., 2025) family? While we cannot necessarily compute the oracle comparator $\pi^*$, we are still able to compare performance conditioned on different numbers of episodes in the thought block. This gives us a sense of whether the cumulative regret grows only very slowly in $T$. To this end, we study the behavior of the DeepSeek-R1-Distill-Qwen-32B model on two datasets: AIME 2024 and a subset from OmniMATH (Gao et al., 2024). In this context, an *episode* is defined as a continuous segment of the model's thought (i.e., text enclosed in between the <think> and </think> markers) uninterrupted by words such as "Wait" and "Alternatively" which break the current flow of logic.

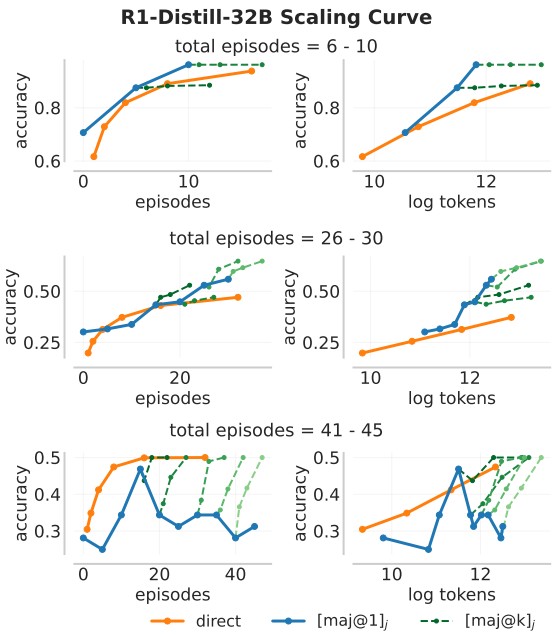

*Figure 3.* ***R1 scaling curve on Omni-MATH subset.*** We compare performance when terminating at the $j$-th episode: blue points represent $[\mathbf{maj@1}]_j$, where the model continues reasoning up to that point before answering; green points represent $[\mathbf{maj@p}]_j$ for $p = 1, 2, 4, 8$, where the model stops at the same point and produces $p$ completions for majority voting. Surprisingly, under the same token and episode budget, early termination with multiple completions (green) often outperforms continued reasoning (blue), suggesting that overthinking can degrade performance.

We report our metrics in terms of the $[\mathbf{maj@p}]_j$ metric, in which we truncate the thought block produced by the LLM to the first $j$ episodes ($\mathbf{z}_{0:j-1}$) and steer it into immediately producing the final solution (without producing more episodes) conditioned on this truncated thought block. We then sample such immediate answers $p$ times and run a majority vote over them to produce a single answer. $p$ and $j$ are variables that parameterize the metric $[\mathbf{maj@p}]_j$ that we measure. We also found that terminating the model's thinking process early requires us to incorporate an intermediate prompt that asks the model to "formulate a final answer based on what it already has" because it has already

spent enough time on this problem[3]. Observing $[\mathbf{maj@p}]_j$ evolve with $j$ tells us if adding more episodes helps the model make meaningful progress and discover the correct solution. We compare this against the **direct** baseline, in which we fine-tune the base model to produce "best guess" responses directly (Qwen2.5-32B-Instruct model based on the same base model; see Appendix I).

**Analysis results.** We plot the average accuracy of the model at different episodes $j = \{0, \cdots, k-1\}$ as a function of the test-time compute (measured in tokens and episodes) and the episode index $j$ in Figure 3. In particular, we average across solutions that contain similar numbers of episodes (total episodes = 6 - 10, 26 - 30, 41 - 45) to demonstrate the relationship between steady improvement and total episodes. We plot the performance of the direct baseline in orange, and the performance of $[\mathbf{maj@1}]_j$ at different $j$ in blue. The dashed green lines branching from the blue curve extend average accuracy at the end of a given episode $j$, or alternatively, $[\mathbf{maj@1}]_j$ (note that maj@1 = average accuracy on the given problem) to $[\mathbf{maj@p}]_j$ for different number $\mathbf{p}$ of solutions given the thinking trace.

**Takeaways.** When provided with a few episodes (top row in Figure 3; 6 - 10), cumulative regret is low and each new episode continuously reduces regret, whereas $[\mathbf{maj@p}]_j$ and the direct baseline grow slower. However, in settings that require more episodes (e.g., 41-45 episodes in the bottom row and more examples in Appendix I), we find that the accuracy (blue line) does not increase with each episode, and sometimes degrades with each subsequent episode generated in the output stream. This illustrates that current training does not quite produce traces that optimize regret swiftly (Figure 3), despite it being possible to minimize regret from intermediate episodes using information present in the model (as indicated by the much better performance of $[\mathbf{maj@p}]_j$ when the total number of episodes $\in [41, 45]$).

***This result is even more surprising because:*** a long trace with multiple sequential episodes should be perfectly capable of implementing the $[\mathbf{maj@p}]_j$ baseline as there is no new knowledge needed to implement this baseline. It should also easily beat the direct baseline, which just reasons in a direct/linear chain and does not perform long CoT reasoning. However, reasoning with sequential episodes loses to both baselines when the solution contains more episodes. Inconsistent progress with many episodes implies poor performance as we scale up test-time compute even further.

---

[3]We discovered that similar statements are used to limit the thinking time of R1 models when it outputs an exceedingly long solution. Following such a statement, R1 would end the thinking block and give a final answer. To make sure that a rather premature trimming of the thought block results in natural terminations and does not alter the model's abilities in a detrimental manner, we manually incorporated a suffix of this sort when computing $[\mathbf{maj@p}]_j$. The exact prompt is shown in Appendix I.

If outcome-reward RL was imbuing the LLM with generalizable test-time scaling, we would expect it to improve consistently.

> **Takeaways: Existing models do not minimize regret**
> - Additional reasoning in models trained with outcome reward RL do not consistently yield a performance improvement, particularly for complex problems that require many episodes.
> - Even when better performance can be achieved by implementing "naïve" strategies such as majority voting on fewer episodes, a long sequential chain of thought is unable to implement those.

## 4. The *MRT* Paradigm

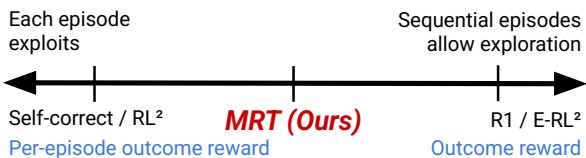

*Figure 4.* ***Explore/exploit spectrum.*** Final reward RL does not reward intermediate episodes encouraging unstructured exploration, whereas SCoRe (Kumar et al., 2024; Qu et al., 2024) constrains each episode based on its outcome reward making it too exploitative. ***MRT*** strikes a balance by assigning an information gain based reward which aims to make progress in a budget-agnostic setting.

We will now develop a fine-tuning paradigm that we call **meta reinforcement fine-tuning (MRT)** that directly aims to learn a budget-agnostic LLM, which makes steady progress. Abstractly, ***MRT*** fine-tunes LLMs to directly optimize (a surrogate to) cumulative regret.

Optimizing outcome reward over a long stream does not incentivize meaningful regret minimization during test-time. As long as the LLM finds *some* arbitrary way to eventually succeed, all intermediate episodes in this rollout will be equally reinforced without accounting for the contribution of every episode towards the eventual success. This is problematic for two reasons: **(i)** we may simply run out of the deployment token budget to discover solutions to hard problems if we are not making progress, and **(ii)** we will waste the token budget on easy problems that could be solved otherwise more efficiently. One way of addressing these issues is to directly optimize for the cumulative regret objective (Definition 2.1). However, this is problematic due to the presence of the optimal comparator policy $\pi^*$, which we do not have access to. The inability to access $\pi^*$ is not new or surprising: even over training of any RL algorithm, we do not have access to the comparator policy for minimizing cumulative regret. The difference here is that ***this cumulative regret is not measured over training steps but rather on test-time token output on a given test query*** (see Figure 1(b), where the regret corresponds to the red area). As a result,

in this section, we come up with a surrogate objective that trains the LLM to implement a regret-minimizing strategy when deployed. This should allow us to strike a balance between spending tokens on exploration and exploitation at test time (Figure 4); exploration in the sense of trying new approaches, verifying prior answers, running majority voting and exploitation in the sense of committing to simplifying an expression following a given plan.

### 4.1. Surrogate Objectives for Minimizing Regret

The regret (Definition 2.1) cannot be directly optimized since the optimal comparator $\pi^*$ is not known. Our main idea is that we can minimize cumulative regret over episodes produced by $\pi$ if we maximize "progress" of policy $\mu$ as more episodes are produced. To see why intuitively, we provide a simple analogy with a multi-armed bandit learning problem where we must learn to discover the optimal arm and rewards are not noisy. There are two behaviors that we must tradeoff to minimize cumulative regret in a bandit problem: **1)** stumbling upon promising but risky arms, and **2)** continuing to exploit the best arm known so far. In either case, each subsequent arm pull should lead to non-zero and ideally positive improvement in the performance of an "exploitation" policy that aims to simply produce the best guess estimate of the optimal arm given the episodes so far.

We use this framework to build a simple surrogate objective. The episodes $\mathbf{z}_{0:k}$ are analogous to "arm pulls" in our setting, with the meta-prover policy $\mu$, serving the role of the policy which aims to estimate best arm. We can hope to see regret minimized as long as the meta-prover $\mu$ makes progress, i.e., $J_r(\mu(\cdot|\mathbf{x}, \mathbf{z}_{0:j}))$ increases with more episodes $\mathbf{z}_j$. Note that this does not mean that each subsequent episode $\mathbf{z}_j$ must itself contain a better solution like SCoRe (Kumar et al., 2024) or RISE (Qu et al., 2024), but only that it should ideally increase the probability that $\mu$ arrives at the right answer (Figure 4). Following the formalism in Setlur et al. (2024b), we capture this notion of progress made by $\mu$ via advantage of an episode $\mathbf{z}_i$ under $\mu$.

---

**Definition 4.1** (***Progress***). Given prior context $\mathbf{c}$ and episode $\mathbf{z}_j \sim \pi(\cdot|\mathbf{c})$, and another meta-prover LLM $\mu$ that computes an estimate of the correct response, we define progress made by $\mathbf{z}_j$ as
$$r^{\mu}_{\mathrm{prg}}(\mathbf{z}_j; \mathbf{c}) = J_r(\mu(\cdot|\mathbf{z}_j, \mathbf{c})) - J_r(\mu(\cdot|\mathbf{c})).$$

---

### 4.2. Incorporating Progress as a Dense Reward Bonus

Defining the standard fine-tuning loss function based on the expected final reward attained by the last episode as the following objective, $\ell_{\mathrm{FT}}$:

$$\ell_{\mathrm{FT}}(\pi) := \mathbb{E}_{\mathbf{x}\sim\mathcal{D}_{\mathrm{train}},\mathbf{z}\sim\pi(\cdot|\mathbf{x})}\left[r(\mathbf{x}, \mathbf{z})\right], \qquad (2)$$

we can train the LLM $\pi$ either with the policy gradient obtained by differentiating Equation 2 or with SFT on self-generated data (Singh et al., 2023). We can extend Equa-

tion 2 to incorporate progress, giving rise to the abstract training objective ($\mathbf{c}$ is the sequence of tokens generated so far), $\ell_{\mathrm{MRT}}(\pi; \pi_{\mathrm{old}}) := \ell_{\mathrm{FT}}(\pi) + \alpha \cdot \ell_{\mathrm{PRG}}(\pi)$, where $\ell_{\mathrm{PRG}}(\pi)$ is defined as:

$$\mathbb{E}_{\mathbf{x}\sim\mathcal{D}_{\mathrm{train}}}\left[\textcolor{red}{\sum_{j=0}^{k-1}\mathbb{E}_{\substack{\mathbf{c}_{j-1}\sim\pi_{\mathrm{old}}(\cdot|\mathbf{x})\\\mathbf{z}_j\sim\pi(\cdot|\mathbf{c}_{j-1})}}\left[r^{\mu}_{\mathrm{prg}}(\mathbf{z}_j; \mathbf{c}_{j-1})\right]}\right]. \quad (3)$$

The term in red corresponds to the reward bonus and it is provided under the distribution of contexts $\mathbf{c}_{j-1}$ consisting of prefixes produced by the previous LLM checkpoint, shown as $\pi_{\mathrm{old}}$. The meta prover policy $\mu$ can be any other LLM (e.g., an "-instruct" model which is told to utilize episodes so far to guess the best answer) or the same LLM $\pi$ itself after its thought block has terminated.

Utilizing the previous policy $\pi_{\mathrm{old}}$ in place of the current policy $\pi$ serves dual purpose: **(1)** akin to trust-region methods in RL (Schulman et al., 2015; Peng et al., 2019), it allows us to improve over the previous policy provably, and **(2)** it lends ***MRT*** amenable to a more convenient implementation on top of RL or STaR infrastructure that need not run "branched" rollouts (Kazemnejad et al., 2024), and can use an off-policy or stale distribution of contexts. Prior work (Setlur et al., 2024b) alleviates the need for branched rollouts by training an explicit value function, but often induces errors. Therefore, we opt to use off-policy contexts but provide additional rewards. We also remark that this additional reward can be provided to the segment of tokens spanning a particular episode ("per-episode" reward) or as a cumulative bonus at the end of the entire test-time thinking trace, with alternatives resulting in different variance for the gradient update. Unlike traditional RL that optimizes outcome rewards and recent approaches that provide step-level supervision, MRT aligns with meta-RL by operating at the meta-step (episode) level, assessing progress across complete reasoning trajectories rather than individual actions.

Finally, while this objective might appear similar to that of Setlur et al. (2024b), we crucially note that the progress is not computed over steps appearing within one attempt but rather over episodes. With this abstract objective in place, we now write down concrete instantiations for SFT and RL.

## 5. Practical Instantiations: Dense Rewards for Optimizing Test-Time Compute

We now instantiate ***MRT*** to train an LLM in a way that enables it to learn to use test-time compute effectively and efficiently. We parameterize each episode as a logical thought block enclosed in between the <think> markers, akin to the DeepSeek-R1 model. As shown in Figure 5 (Left), we refer to this as an "open-ended parameterization" since it does not constrain the content of each episode. With this parameterization, we optimize the objective in Definition 3

with STaR (Zelikman et al., 2022) and RL (Shao et al., 2024). With STaR, this involves sampling on-policy traces, followed by behavior cloning the ones that not only succeed under the outcome reward, but also attain high progress. With RL, this involves either explicitly or implicitly adding a reward bonus that corresponds to progress.

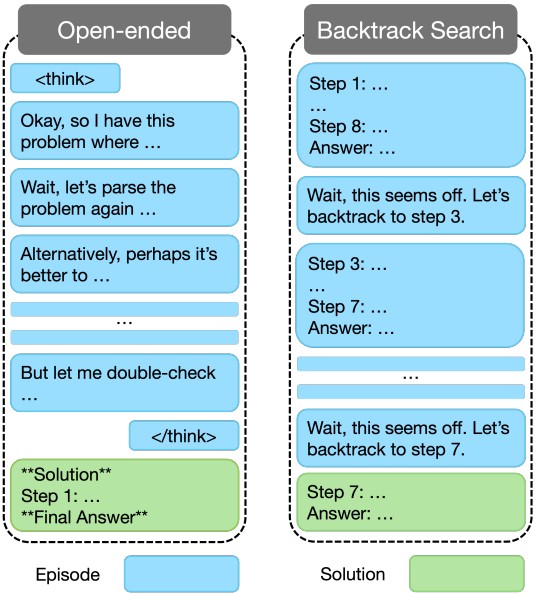

*Figure 5.* ***The two settings we study.*** **Left:** open-ended parametrization. The model uses explicit thinking markers (<think> and </think>) to work through a problem with multiple strategies. **Right:** backtracking search. The model directly solves the problem with a step-by-step solution. In each episode, the model identifies errors at specific steps and backtracks to correct them (returning to step 3, then later to step 7) until reaching the correct answer.

We also study a "backtracking search" parameterization (Figure 5, Right) where the model alternates between full solution attempts and backtracking; details of this approach along with empirical results are provided in Appendix C.

### 5.1. STaR and RL Variants of *MRT*

We build two *MRT* variants that optimize test-time compute via on-policy rollouts and dense progress rewards: one based on **STaR**, the other on **RL**.

The **STaR** variant of *MRT* leverages self-generated roll-outs from the base model $\pi_b$ to create a filtered dataset of high-quality traces for SFT. For each input prompt $\mathbf{x}$, we sample an initial trace $\mathbf{z}$ between <think> tags. We then segment the reasoning trace $\mathbf{z}$ into episodes $\mathbf{z}_0, \mathbf{z}_1, \cdots, \mathbf{z}_n$. The meta-prover policy $\mu$ is implemented as the policy that forcefully terminates the thought block with the "time is up" prompt (Appendix I; used in our analysis) and forcing the model to produce a solution given prefix:

$$\mu(\cdot | \mathbf{x}, \mathbf{z}_{0:j}) \stackrel{\text{def}}{=} \pi_b(\cdot | \mathbf{x}, \mathbf{z}_{0:j}, [\text{time is up}], </\text{think}>) \quad (4)$$

We compute progress $r_{\text{prg}}^{\mu}(\mathbf{z}_j, \mathbf{x})$ according to Definition

4.1. Now, we filter for episodes $\mathbf{z}_{0:j}$ that satisfy two criteria: **(1)** they achieve maximum progress, i.e., $j = \arg\max_j \sum_{k=0}^{j} r_{\text{prg}}^{\mu}(\mathbf{z}_k; \mathbf{c}_{k-1})$, where $\mathbf{c}_{k-1} \equiv (\mathbf{x}, \mathbf{z}_{0:k-1})$ and **(2)** they eventually succeed, i.e., if $\mathbf{y} \sim \mu(\cdot | \mathbf{x}, \mathbf{z}_{0:j})$ then $r(\mathbf{x}; \mathbf{y}) = 1$. And finally, we run SFT on these traces, and repeat the process for multiple iterations.

The **RL** variant of *MRT* using online RL methods (e.g., GRPO (Shao et al., 2024) or PPO (Schulman et al., 2017)) to optimize progress-based rewards. For each episode, we compute prefix rewards using the meta-prover $\mu$ (Equation 4, Figure 6). The model then samples multiple on-policy roll-outs conditioned on this prefix, evenly divided between continuing to reason and terminating right after the prefix of the thinking trace and producing the best-guess solution. During training, we optimize the reward defined in Equation 3 rather than just the binary outcome reward. While this procedure can be implemented with episode-specific reward bonuses or a single progress adjusted reward, we opt for the latter approach due to its plug-and-play nature in current outcome-reward RL implementations.

## 6. Experimental Evaluation

We now evaluate how effectively *MRT* optimizes test-time compute—focusing on maximizing accuracy while minimizing compute. We discuss our main results below, then compare the efficiency of *MRT* against other prior methods, and finally end with ablation experiments studying the relationship between token budget and progress.

### 6.1. Experimental Setup

We use *MRT* to fine-tune base models that can already produce traces with <think> markers. For the STaR variant, we use DeepSeek-R1-Distill-Qwen-7B and 1.5B, fine-tuned on 10K randomly sampled problem-solution pairs from NuminaMath (Li et al., 2024) and estimate the progress bonus for backtracking by rolling out each prefix 20 times. Here, we compare *MRT*, which incorporates progress as a bonus, versus vanilla STaR which only uses outcome reward. For the RL variant, we utilized DeepSeek-R1-Distill-Qwen-1.5B and DeepScaleR-1.5B-Preview as base models (omitting the 7B model due to higher training compute requirements), where we compare *MRT* with outcome-reward RL (vanilla GRPO (Shao et al., 2024)). We finetuned DeepSeek-R1-Distill-Qwen-1.5B with *MRT* on 4,000 NuminaMath problems, while DeepScaleR-1.5B-Preview, which had already undergone one round of outcome-reward RL finetuning on 40K MATH problem-answer pairs, was finetuned only on 919 AIME problems from 1989-2023. We also compare *MRT* to an RL approach that explicitly penalizes the token length. The average number of tokens in a response on evaluation prompts is around 8k, therefore, we fine-tune with a 16K maximum token budget and evaluate at the same budget. More details are outlined in Appendix G.1, and a complete set of hyperparameters can be found in Appendix G.2.

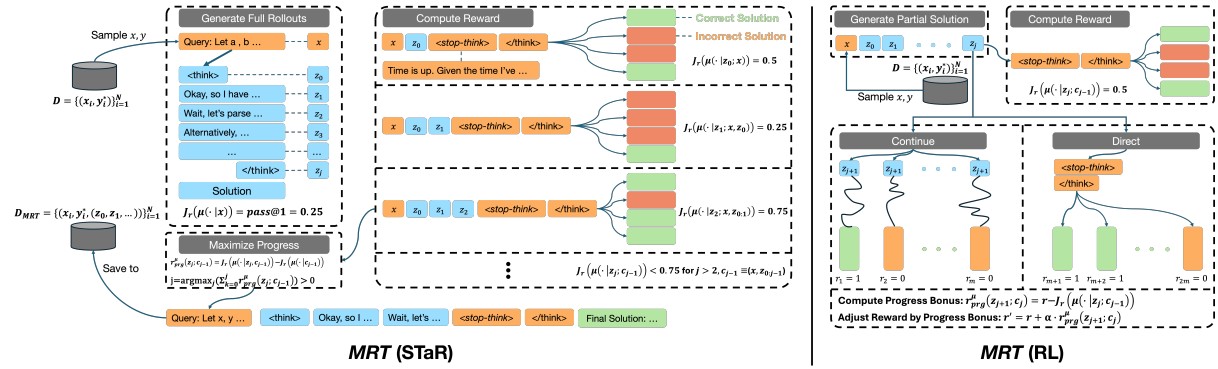

*Figure 6.* **MRT implementation**. **Left:** The **STaR** variant begins by generating a complete rollout for each query $\mathbf{x}$ sampled from dataset $\mathcal{D}_{\text{train}}$. Then, **MRT** segments thinking traces into distinct episodes $\mathbf{z}_j$ akin to our analysis in Section 3. For each prefix $\mathbf{z}_{0:j}$, we estimate reward $J_r(\mu(\cdot|\mathbf{z}_{0:j}, \mathbf{x}))$ by evaluating the average accuracy of solutions produced after terminating the thought block at this prefix. After computing rewards across all prefixes, we calculate progress $r_{\text{prg}}^\mu(\mathbf{z}_{0:j}; x)$ using Definition 4.1. The STaR variant selectively retains only reasoning traces that maximize progress and are also followed by correct solutions once thinking terminates. **Right:** The **RL** variant initiates by generating a partial rollout for each query $\mathbf{x}$ sampled from $\mathcal{D}_{\text{train}}$, terminating after a random number of episodes. Then it generates $m$ on-policy rollouts that terminate reasoning at the prefix and immediately produce final solutions as well as rollouts that continue reasoning. Normalizing rewards across this set of traces allows us to implicitly compute the progress bonus. Finally, we update the policy with an aggregation of this dense reward and the final 0/1 outcome reward.

| Base model + Approach | AIME 2024 | AIME 2025 | AMC 2023 | MinervaMATH | MATH500 | Avg. |
|---|---|---|---|---|---|---|
| **DeepScaleR-1.5B-Preview** | 42.8 | 36.7 | 83.0 | 24.6 | **85.2** | 54.5 |
| outcome-reward RL (GRPO) | 44.5 (+1.7) | 39.3 (+2.6) | 81.5 (−1.5) | **24.7** | 84.9 | 55.0 (+0.5) |
| length penalty | 40.3 (−2.5) | 30.3 (−6.4) | 77.3 (−5.7) | 23.0 | 83.2 | 50.8 (−3.7) |
| **MRT** (Ours) | **47.2** (+4.4) | **39.7** (+3.0) | **83.1** (+0.1) | 24.2 | 85.1 | **55.9** (+1.4) |
| **R1-Distill-Qwen-1.5B** | 28.7 | 26.0 | 69.9 | 19.8 | 80.1 | 44.9 |
| outcome-reward RL (GRPO) | 29.8 (+1.1) | 27.3 (+1.3) | 70.5 (+0.6) | 22.1 | 80.3 | 46.0 (+1.1) |
| **MRT** (Ours) | **30.3** (+1.6) | **29.3** (+3.3) | **72.9** (+3.0) | **22.5** | **80.4** | **47.1** (+2.2) |

*Table 1.* **Pass@1 performance of RL-trained MRT models on various math reasoning benchmarks.** We compare **MRT**, outcome-reward RL (GRPO), and length-penalized RL against strong base models. **MRT** consistently outperforms all methods, achieving state-of-the-art results in its size class. **MRT** *leads to a 2-3x improvement in accuracy over the base model compared to that of outcome-reward GRPO.* Note that both base models are already trained with RL on a potentially a larger superset of prompts, or distilled from RL trained models, and thus we should expect the gains from any subsequent fine-tuning to be small in absolute magnitude. Despite this, we observe a statistically significant and systematic gain with **MRT**, which is **2 − 3×** of the gain from outcome-reward training.

## 6.2. Results for *MRT*

Following the protocol in Luo et al. (2025), we report the pass@1 performance of outcome-reward RL and **MRT** on multiple math reasoning datasets: AIME 2025, AIME 2024, AMC 2023, MinervaMATH, and MATH500, using 20 samples per problem to reduce noise due to limited size.

As shown in Table 1, **MRT** outperforms training on the same dataset without the dense reward bonus. We additionally make a number of interesting observations and draw the following takeaways: *(a) State-of-the-art results.* To the best of our knowledge, our models fine-tuned on top of the DeepScaleR-1.5B-Preview base model achieve state-of-the-art performance for their size. The absolute performance gains are small because we train on top of distilled or already RL-trained base models. However, *the relative performance improvement from using* **MRT** *is about 2-3x compared to the performance improvement obtained from running outcome-reward RL (GRPO). (b) Better out-of-distribution robustness.* When fine-tuned on a narrow dataset of AIME

problems with the DeepScaleR-1.5B model, **MRT** not only attains better performance on AIME 2024 and AIME 2025 evaluation sets (which is perhaps expected), but **MRT** also preserves performance on the AMC 2023 dataset that is somewhat out-of-distribution compared to outcome-reward RL. *(c) Larger gains with weaker models and broader training data.* The gains in performance are further exaggerated on the DeepSeek-R1-Distill-Qwen-1.5B model in comparison, since the DeepScaleR base model is already trained with RL, whereas the latter is not.

We also evaluate DeepScaleR-1.5B with an explicit length penalty to improve token efficiency, following Arora & Zanette (2025). Consistent with their findings, we observe that the length penalty reduces pass@1 accuracy.

## 6.3. Token Efficiency of *MRT*

So far we have seen that **MRT** can improve performance beyond standard outcome-reward RL in terms of pass@1 accuracy. Next, we try to evaluate whether **MRT** (RL) also leads to an improvement in the token efficiency needed to

solve these problems. To plot token efficiency, we train the model with a 16K context window and compute maj@K on multiple reasoning and solution traces sampled from the LLM. Plotting maj@K against token usage provides us with an estimate of the model performance *per token*. As shown in Figure 7, in both STaR and RL settings, **MRT** outperforms the base model by an average of **5%** accuracy given the same number of tokens on AIME 2024. Moreover, **MRT** (RL) requires **5x** fewer tokens on AIME 2024 and around **4x** fewer tokens on MATH 500 to achieve the same performance as the base model (DeepSeek-R1 distilled Qwen-1.5B model in this example). In a similar vein, **MRT** improves over outcome-reward RL by **1.2-1.6x** in token efficiency. These results demonstrate that **MRT** significantly improves token efficiency while maintaining or improving accuracy. We also evaluated training 7B base models **MRT** (STaR). We present these results in Appendix H.1. A detailed analysis of the computational cost trade-offs, showing only **1.01×** and **1.08×** overhead in FLOPs compared to STaR and GRPO respectively, is provided in Appendix E.

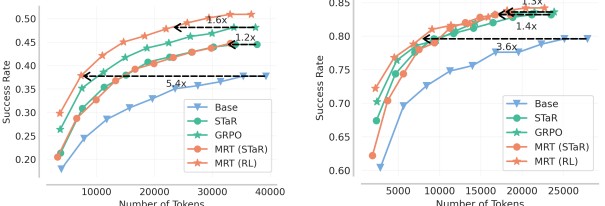

*Figure 7.* **MRT (RL and STaR) results on DeepSeek-R1-Distill-Qwen-1.5B**. We plot maj@k for k = 1, 2, ..., 10 on AIME 2024 (left) and MATH500 (right). The orange lines correspond to **MRT** and the green lines correspond to outcome-reward training, with ★ denoting RL and ● denoting STaR / SFT training.

## 6.4. Ablation Studies and Diagnostic Experiments

Next, we perform controlled experiments to better understand the reasons behind the efficacy of **MRT**. We aim to answer the following question: **Do MRT (RL) and MRT (STaR) reduce cumulative regret and make more progress compared to outcome-reward RL and STaR?** In the main text, we focus on this core question by analyzing regret reduction as a function of token budget. Additional diagnostic experiments examining the relationship between token length and progress—including how length evolves over training and how curriculum strategies influence performance—are presented in Appendix D.

### 6.4.1. PROGRESS MADE BY **MRT** COMPARED TO OUTCOME-REWARD TRAINING

We measure the regret from Definition 2.1 against an optimal "theoretical" policy $\pi^*$ that achieves perfect accuracy in one episode. While Definition 2.1 measures regret $\Delta_k^{\mu}$ as a function of the number of episodes $k$, to fairly compare different fine-tuning algorithms, we instead reparameterize regret to be a function of token budget $C_0$ for this study. Since traces from different algorithms can differ in the num-

ber of episodes, cumulative regret per token provide a more apples-to-apples comparison of progress. Specifically, we measure the scaling curve (blue curve in Figure 1) and cut it off at varying budgets of $C_0$. We then measure the area ratio between the scaling curve at different values of $C_0$ and the constant oracle performance of 1.0 (visually depicted as the shaded red area in Figure 1). Finally, we report this regret normalized by $C_0$ in Figure 8.

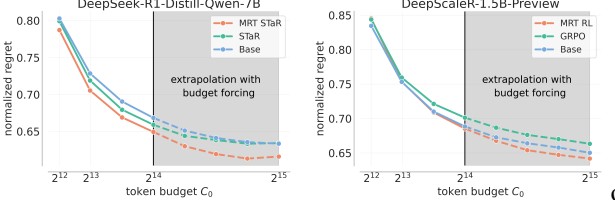

*Figure 8.* **Normalized regret of different algorithms at different deployment @token budgets** $C_0$. The first four points correspond to budgets of 4096–16384 tokens; the next four (dashed) are extrapolations to 20480–32768 using the budget-forcing method from s1 (Muennighoff et al., 2025). The left plot shows the STaR variant of **MRT**, and the right shows the RL variant on DeepScaleR-1.5B-Preview, both evaluated on AIME 2025. **MRT** consistently achieves the lowest normalized regret, even as outcome-reward methods plateau or regress at higher budgets.

A low and steadily decreasing normalized regret indicates the "red" area in Figure 1 narrows as token usage grows. Empirically, we see in Figure 8 that the normalized regret for **MRT** decreases faster compared to both the base model and outcome-reward RL when the total token budget $C_0 \leq 16384$, the token budget used for training.

In Figure 8, we also include token budgets that extrapolate beyond training budget, shown in the dashed lines. To do so, we force the model to continue thinking using the budget forcing approach of Muennighoff et al. (2025). Even in extrapolation, **MRT** continues to have the lowest normalized regret, indicating better progress at larger budgets. We present a detailed version of this study in Appendix K.

## 7. Conclusion

We introduce **MRT**, a framework for optimizing test-time compute in LLMs via meta reinforcement learning. By minimizing cumulative regret—a measure of how efficiently a model uses its available compute—**MRT** overcomes the limitations of outcome-reward RL, which fails to reward partial progress and often misallocates test-time tokens. **MRT** introduces a dense reward bonus that quantifies incremental progress during generation and enables learning policies that better utilize the test-time budget. Empirically, this leads to stronger performance, lower regret, and more effective extrapolation beyond the training budget. We believe this formulation opens several exciting directions for future work, including improvements in meta-prover design, base model diversity, rollout strategies, and systematic evaluation under compute-matched regimes. We elaborate on these open questions and challenges in Appendix F.

# Impact Statement

This paper presents work whose goal is to advance the field of Machine Learning. There are many potential societal consequences of our work, none which we feel must be specifically highlighted here.

# Acknowledgements

We thank Max Sobol Mark, Seohong Park, Bhavya Agrawalla, Zheyuan Hu, Fahim Tajwar, Kelly He, So Yeon Min, Martin Q. Ma, Ian Wu, Charlie Snell, Rafael Rafailov, Anikait Singh, Yi Su, Kwanyoung Park, Paria Rashidinejad, Tianjun Zhang, Michael Luo, Aditi Raghunathan, Andrea Zanette, Max Simchowitz, and Rishabh Agarwal for their feedback and informative discussions. This work was supported by the Office of Naval Research under N00014-24-12206, OpenAI SuperAlignment grants, and used the Delta system at the National Center for Supercomputing Applications through CIS240761 and CIS240253, supported by the NSF. AK and AS would especially like to thank Lunjun Zhang for informative discussions on the meta reinforcement learning formulation. AS is thankful for the generous support of JP Morgan AI PhD Fellowship. RS is supported in part by ONR grant N00014-23-12368.

This work was built upon TRL (von Werra et al., 2020) and Open-R1 (Face, 2025). Our research would not be possible without these open-source projects. We would like to express special thanks to Quentin Gallouédec, Kashif Rasul, and the rest of the Hugging Face team for their invaluable guidance, technical insights, and continuous support with Open-R1 and TRL. This expertise significantly sped up the development of our methods. We also thank Michael Luo and Tianjun Zhang for sharing the DeepScaleR checkpoints for analysis.

# Author Contributions

YQ led the experimentation in the final paper, with support from LT, EB, and AS. MY led the case study, probing experiments, and scaling analysis, with support from YQ and AS. LT and EB ran the large-scale experiments, and helped with data collection and evaluations. The development of the final *MRT* algorithm and ablation experiments were done by YQ, with inputs from AK and RS. YQ led the infrastructure development with support from LT. YQ, MY, AS, and AK wrote the manuscript, with input from all co-authors. AK, AS, and RS advised on the overall direction and supervised the project.

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

# Appendices

## A. Related Work

**Scaling test-time compute.** Earlier works (Wu et al., 2024; Welleck et al., 2024) scale up test-time compute by training separate verifiers (Setlur et al., 2024b; Chow et al., 2024) for best-of-N (Cobbe et al., 2021) or beam search (Beeching et al., 2024), which can be more optimal than scaling data or model parameters (Snell et al., 2024; Jones, 2021). Building on this, recent works (Gandhi et al., 2024; Moon et al., 2024) train LLMs to "simulate" in-context test-time search by fine-tuning on search traces. However, gains from such approaches are limited since fine-tuning on search traces that are unfamiliar to the base model can lead to memorization (Kumar et al., 2024; Kang et al., 2024; Setlur et al., 2024a). To prevent this in our setting, we apply a warmstart procedure before running on-policy STaR/RL.

**Reasoning with long chains of thought (CoT).** RL with outcome rewards has shown promise for finetuning LLMs to produce long CoTs that can search (Lehnert et al., 2024), plan (Yao et al., 2023), introspect (Qu et al., 2024) and correct (DeepSeek-AI et al., 2025; Kimi-Team, 2025). More recently, several works have considered adding length penalties to the outcome reward objective to discourage length for easier problems (Arora & Zanette, 2025) and encourage length for harder problems (Yeo et al., 2025b; Ye et al., 2025a). However, recent work has shown that length may not have a direct correlation with accuracy (Zeng et al., 2025; Liu et al., 2025; Luo et al., 2025), and that existing long CoT models tend to use too many tokens (Chen et al., 2024). In our work, we tie this inefficiency to the inability of outcome-reward RL to learn to output solutions that make steady progress. Similar to our approach, concurrent works also leverage dense rewards. For example, (Cui et al., 2025), which maximizes the likelihood of generating successful traces given a partial solution, and (Ye et al., 2025a), which obtains the exploration bonus from a length penalty or an LLM judge. However, the dense reward design in *MRT* is inspired by regret minimization and does not require an LLM judge. There have also been efforts to distill the traces generated from existing reasoning models via SFT (Muennighoff et al., 2025; Ye et al., 2025b; Team, 2025b;a), however, these are orthogonal to our work which focuses on improving RL directly. In addition, recent work shows that RL-trained policies scale test-time compute better than SFT (Setlur et al., 2025).

**Meta RL.** We formulate optimizing test-time compute as a meta RL problem (Beck et al., 2023; Gupta et al., 2018b;a). Concurrently, a recent survey (Xiang et al., 2025) posits "how-to-think" with meta chain-of-thought as a promising direction for training the next frontier of reasoning models. In fact, prior work in RL (Ghosh et al., 2021; Rakelly et al., 2019) shows that it is *necessary* to solve a meta RL problem to effectively generalize to unseen initial contexts (*i.e.*, new problems), with a little bit of interaction (*i.e.*, initial episodes or attempts). Most work in meta RL (Finn et al., 2017a; Agarwal et al., 2019; Mendonca et al., 2019) differs in the design of the adaptation procedure. *MRT* is closest to meta RL methods that use in-context histories (Duan et al., 2016; Stadie et al., 2019), but differs in the design of rewards, striking a balance between E-RL$^2$ (Stadie et al., 2019) that does not reward all but the last episode (only exploration), and RL$^2$ (Duan et al., 2016) that rewards each episode (only exploitation).

## B. Preliminaries and Background

**Problem setup.** Our goal is to optimize LLMs to effectively use test-time compute to tackle difficult problems. We assume access to a reward function $r(\mathbf{x}, \cdot) : \mathcal{Z} \mapsto \{0, 1\}$ that we can query on any output stream of tokens $\mathbf{z}$. For example, on a math problem $\mathbf{x}$ with token output stream $\mathbf{z}$, reward $r(\mathbf{x}, \mathbf{z})$ can check if $\mathbf{z}$ is correct. We are given a training dataset $\mathcal{D}_{\text{train}} = \{(\mathbf{x}_i, \mathbf{y}_i^*)\}_{i=1}^N$ of problems $\mathbf{x}_i$ and oracle solution traces $\mathbf{y}_i^*$ that ends in the correct answer. Our goal is to use this dataset to train an LLM, which we model as an RL policy, $\pi(\cdot|\mathbf{x})$. We want to train LLM $\pi$ to produce a stream of tokens $\mathbf{z}$ on that achieves a large $r(\mathbf{x}, \mathbf{z})$ on test problem $\mathbf{x} \sim \mathcal{P}_{\text{test}}$.

**Meta RL primer.** RL trains a policy to maximize the reward function. In contrast, the meta RL problem setting assumes access to a distribution of tasks with different reward functions and dynamics. The goal in meta RL is to train a policy on tasks from the training distribution such that it can do well on the test task. We do not evaluate this policy in terms of its zero-shot performance, but let it adapt by executing "adaptation" episodes at test time. Most meta RL methods differ in the design of this adaptation procedure (e.g., in-context RL such as RL$^2$ (Duan et al., 2016), explicit training (Finn et al., 2017b), and latent inference (Rakelly et al., 2019)).

## C. MRT with the Backtracking Search Parameterization

In addition to the open-ended parameterization discussed in the main text, we explore a more structured approach to episode parameterization that we call "backtracking search". In this setting, we design episodes to alternate between: **(1)** an attempt

to solve the problem, and **(2)** an attempt to discover errors in the preceding attempt, followed by determining an appropriate step to backtrack to. This parameterization explicitly encourages the model to develop error detection capabilities and strategic backtracking, without the use of any <think> markers. Note that the use of no specific <think> marker, and the requirement for each alternate episode to end in some estimate of a solution makes this parameterization be substantially restricted compared to the open-ended setting. That said, this structural constraint of alternating between generation and verification enables us to extrapolate indefinitely by simply filling the context window with the last few related episodes and letting the model run on these. We refered to this as a "sliding window" based linearized evaluation in the main text.

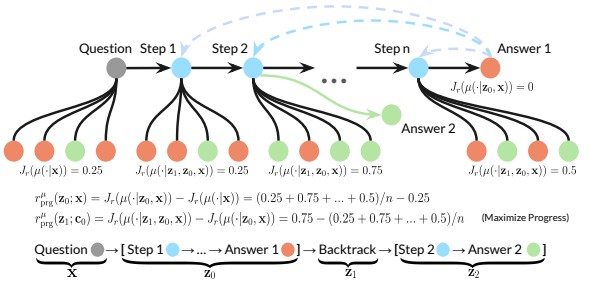

*Figure 9.* ***On-policy rollout*** generation for ***MRT*** in the backtracking search setting. ***MRT*** allows the model to learn to backtrack ($z_1$) and generate the corrected attempt ($z_2$) with a progress-adjusted reward.

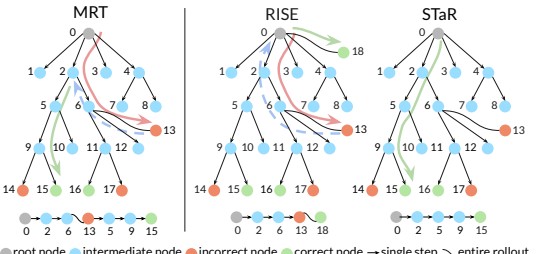

*Figure 10.* ***Different data construction schemes*** for obtaining warmstart SFT data for the backtracking search setting. ***MRT*** traverses two paths with the shared prefix, making use of backtracking, which RISE style approaches.

## C.1. STaR and RL Variants of *MRT* in the Backtracking Search Setting

In this setting, episodes explicitly alternate between generation a solution trace and explicitly implementing a process to implement a form of error correction and backtracking procedure (Figure 5). Concretely, given an initial response $z_0 \sim \pi_b(\cdot|x)$, the subsequent episode $z_1$ is a backtracking episode where the model identifies errors in $z_0$, followed by a corrected attempt $z_2$. Similar to the open-ended setting, in the backtracking search setting, the STaR variant filters on-policy traces (generation of on-policy data depicted in Figure 9) based on **(1)** correctness of $z_2$, i.e., $r(x; z_2) = 1$, and **(2)** high progress backtracks, as measured by a large value of $r_{\text{prg}}^\mu(z_j; c)$. The RL variant follows a similar principle but directly optimizes the progress-adjusted reward rather than the binary outcome, ensuring backtracking leads to meaningful improvements. Finally, we note that although we only train the LLM to optimize for one backtrack, one can run several rounds of backtracks iteratively.

## C.2. Initialization with Warmstart SFT

For the backtracking search setting, we found that base pre-trained LLMs lacked the ability to sample meaningful backtracking operations due to low coverage over such behavior in the pre-training data. This inability to sample backtracks at all, will severely inhibit learning during RL and STaR that rely on self-generated rollouts. Therefore, before running ***MRT*** in the backtracking setting, we had to run an initial phase of "warmstart" supervised finetuning (SFT) to imbue the LLM with a *basis* of backtracking behavior. To do so without human supervision, we generated multiple solution traces by running beam search against the 0/1 outcome reward on every training problem, using rollouts to replace a process reward model (PRM) (Snell et al., 2024). We then generated SFT traces by traversing this tree using a number of heuristics (see Figure 10).

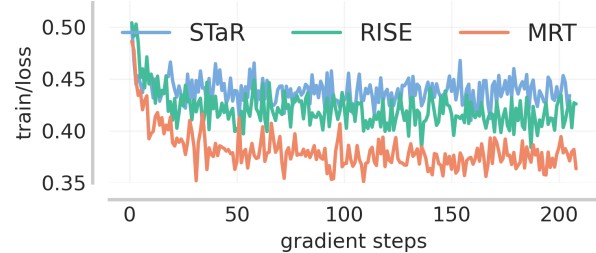

*Figure 11.* ***Training loss*** for warmstart SFT on multiple data configurations: random stitching ("RISE" (Qu et al., 2024)), STaR ("rejection sampling"), and our warmstart SFT data ("Backtrack"). A lower loss implies ease of fitting this data.

We found that backtracking to nodes in the prefix of an attempt that attain a high estimated success rate, followed by completing the solution from there on, resulted in an SFT dataset that was easy to fit without memorization, when normalized for the same token budget. On the other hand, SFT datasets generated by stitching arbitrary incorrect solutions from the beam search tree with a correct solution (e.g., RISE) and direct answer traces were both harder to fit as evidenced by the trend in the training loss in Figure 11. Warmstart SFT was not needed for open-ended parameterizations from R1-distilled checkpoints.

## C.3. Progress Made by MRT Compared to Outcome-Reward Training

We plot the histograms of the progress estimates (Definition 4.1) on episodes obtained by running evaluation rollouts from ***MRT***. We compare them with the progress made by outcome-reward training in Figure 12. Observe that ***MRT*** exhibits a net positive and higher progress over the backtracking episode compared to RISE and outcome-reward RL respectively. This corroborates the idea that ***MRT*** does enhance the progress made by the algorithm.

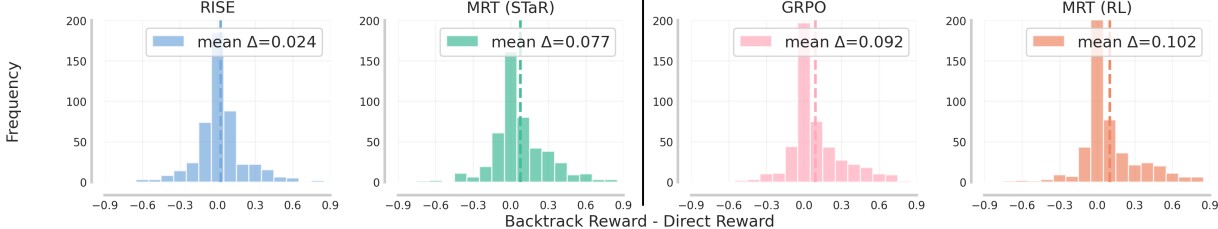

Figure 12. ***Progress histograms in the backtracking search setting*** over the backtracking episode for RISE and ***MRT*** (STaR) on the left and GRPO and ***MRT*** (RL) on right, computed on the evaluation set. In each case, using reward values prescribed by ***MRT*** amplifies information gain on the test-time trace, enabling it to make consistent progress.

## C.4. Linearized Evaluations in the Backtracking Search Setting

Recall that in this setting the model is constrained to producing a solution followed by explicit error detection followed by a revision (Figure 5). The details for how the method is implemented is shown in Appendices G.1 and G.3. When training with ***MRT***, we used Llama-3.1-8B and 3B base models. To generate the training data, we use 20K ranomly-sampled question-solution tuples from the NuminaMath dataset, and sample responses and backtracks from a Llama-3.1-8B model for a "warmstart" SFT phase before running RL training. Our evaluation uses AIME problems from 1989-2023 as a challenging hold-out dataset, where Llama-3.1 8B achieves pass@10 $\approx 30\%$, much lower than the $\approx 60\%$ on NuminaMATH training set. We compare to outcome-reward RL, but also compare ***MRT*** (STaR) to RISE (Qu et al., 2024), a self-correction approach which does not utilize backtracking but just revises the solution.

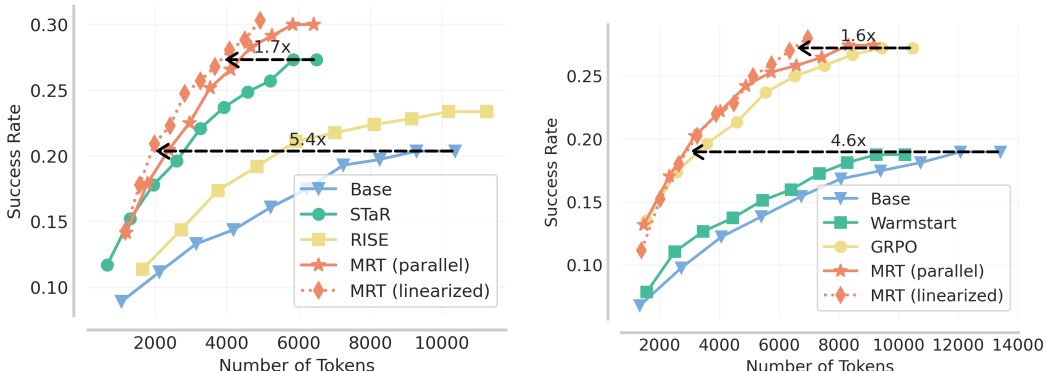

Figure 13. **Left: *MRT* (STaR) with 8B base**. We plot maj@K performance of models on AIME for K $\in [1, 10]$ against the total tokens spent. We also run linearized search (dashed line) for MRT (rest are parallel). **Right: *MRT* (RL) with 3B base**. Similarly to the left plot, we report maj@K against the total tokens spent.

**Evaluation protocol.** Following prior work (Qu et al., 2024), in this setting, we evaluate ***MRT*** in two modes: **(i)** *parallel mode*: sampling $N$ independent three-episode traces (generate-backtrack-revise) per problem and computing maj@N for evaluation; and **(ii)** *linearized mode:* running $N$ sequential episodes of backtracking in a sliding window fashion while retaining the last 2048 tokens, which allows for generating very long but coherent outputs, much longer than the allowed context length for training. Note that this kind of a sliding window evaluation was not possible for the open-ended parameterization, but the use of a more rigid definition of episodes and the Markov property allows us to extrapolate far beyond here.

**Results for *MRT* (STaR).** We first evaluate the STaR variant of ***MRT*** when fine-tuning a Llama-3.1-8B model. As shown in Figure 13 (left), ***MRT*** achieves the highest test-time efficiency in both evaluation modes (parallel in solid lines; linearized in dashed lines) and improves efficiency by over 30% in the linearized evaluation mode. While RISE (Qu et al., 2024)–which does not explicitly model backtracking and does not account for progress–also improves performance, it does so inefficiently, trailing behind ***MRT*** in both the peak performance attained and the number of tokens needed to attain this performance.

**Results for *MRT* (RL).** Finally, we evaluate the RL variant of *MRT* on top of GRPO (Shao et al., 2024) when fine-tuning a 3B model after warmstart SFT (Section C.2). Figure 13 (right) shows that *MRT* (RL) improves linearized efficiency by reducing tokens by 1.6x compared to outcome-reward GRPO.

## D. Evolution of Length and Progress over Training

Finally, we study the relationship between progress and response length, which is believed to be a crucial enabling factor behind the recent results from DeepSeek (DeepSeek-AI et al., 2025) and others (Kimi-Team, 2025). We are interested in understanding: **a)** how does length evolve during training with *MRT* and outcome-reward RL, over an i.i.d. prompt distribution? And **b)** Can the benefits of increasing output token budget be explained by implicitly improving progress? We present results to answer these questions below.

**a) Evolution of completion length during training.** As shown in Figure 14, we find that in general, the average completion length roughly oscillates around a given range of ∼5000 tokens during training with both *MRT* (RL) and GRPO on the AIME dataset (same setup as Table 1). We also note that compared to GRPO, *MRT* slightly reduces length (*i.e.*, the orange curve generally falls below the green curve), which aligns with our expectation that optimizing for progress should lead to some amount of reduction in token length (consistent

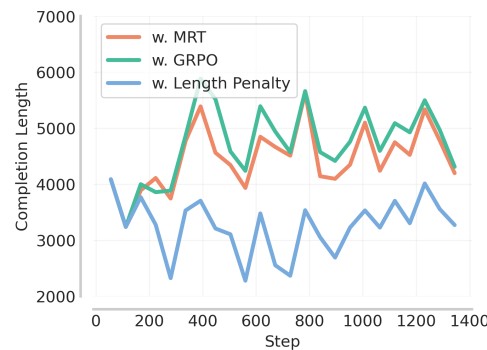

*Figure 14.* ***Evolution of length during RL training.*** Length largely oscillates around similar values for the most part of training, after an initial increase from the initialization length.

with Figure 7). However, this decrease in response length is not as large as the one seen from an explicit length penalty, which reduces length at the cost of worse performance as shown in Table 1. We observe a similar result in the backtracking setting in Figure 20.

**b) Progress explains the benefits of increasing output token budget during training.** Despite the supposed gains from running RL training with a large output budget right from the beginning (DeepSeek-AI et al., 2025; Kimi-Team, 2025), several analyses and reproduction studies (Yeo et al., 2025a; Liu et al., 2025; Zeng et al., 2025; Luo et al., 2025) have found that that training at higher budgets (*e.g.*, a budget of 16K for AIME evaluations) results in inefficient use of compute. Concurrent work, Luo et al. (2025), finds that a more performant approach is to instead initialize RL training with a smaller output token budget of 8K tokens and then expand this budget to 16K after training for some time. This raises the question: what benefits does a "curriculum" over output token budget provide in this setup? In the following discussion, we argue that the benefits of such a curriculum can be explained by increased progress or lower cumulative regret in our formulation.

We start by revisiting the trend in completion length and performance observed by DeepScaleR (Luo et al., 2025) in Figure 15. Observe that when fine-tuning with an 8K context window (training steps 0 to 1000), performance increases while length reduces, implying that an increase in length is not necessary for performance to go up. More interestingly, this trend also indicates that the LLM makes better progress on average during this phase. In particular, the change in accuracy per token/episode is higher than when the token budget is 16K in the next phase, in which both performance and length increase. To corroborate this claim, we compute the normalized regret in Figure 16. We observe that the 8K checkpoint indeed attains a lower regret, meaning each episode in this LLM makes more progress compared to the model trained on 16K.

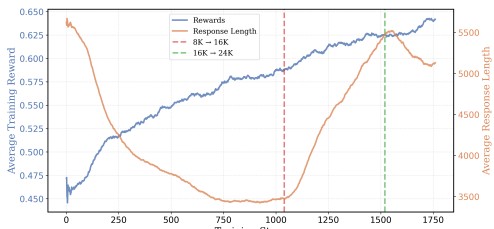

*Figure 15.* (Source: (Luo et al., 2025)) ***DeepScaleR's average response length and training rewards as training progresses.***

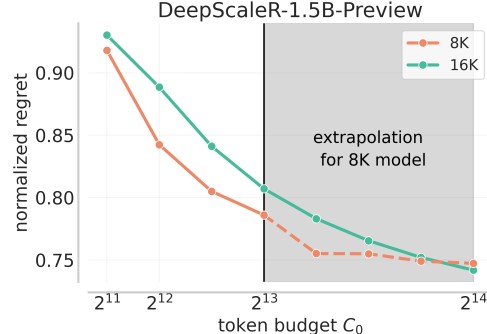

*Figure 16.* ***Regret for 8K and 16K DeepScaleR checkpoints at different budgets*** $C_0$. For budgets beyond 8192, we calculate the normalized regret of the 8K checkpoint by extrapolating it with budget forcing. At nearly all budgets, the 8K checkpoint shows lower normalized regret, indicating better progress.

In fact, even when we extrapolate the budget for the 8K checkpoint to 16K evaluation tokens via budget forcing, we attain a normalized regret similar to the subsequent checkpoint obtained after growing the budget to 16K tokens. Concurrent work (Yeo et al., 2025a; Luo et al., 2025) observes that training with a length curriculum achieves better performance than training with a budget of 16K from scratch. So, the first phase of training on a smaller (8K) token budget results in **a)** higher progress (lower cumulative regret) and **b)** better performance than training with a larger context length, because the latter does not explicitly maximize progress. All of this implies that progress is critical towards driving the benefits of long lengths.

***Our main takeaway*** is that while training with long completion length alone does not always encourage steady progress Liu et al. (2025); Yeo et al. (2025a), some form of an iterative budget curriculum or the dense reward bonus in ***MRT*** can optimize progress. Similar multi-stage training strategies were found critical by prior work training for self-correction (Qu et al., 2024; Kumar et al., 2024). Of course, it is an open question as to how we should instantiate such an iterative training procedure to maximize progress more directly.

> **Insights from ablations: Progress vs length in optimizing test-time compute**
>
> Simple length penalties improve token efficiency but ultimately sacrifice peak performance. Using dense rewards in ***MRT*** increases performance while slightly reducing length, which is a net positive on token efficiency. Existing approaches for using curricula over the training budget or multi-stage training serve as an *implicit* way to encourage progress during RL training.

## E. Total Computation Cost of *MRT*

We performed a detailed analysis of the computational costs associated with our proposed MRT method compared to classical approaches like STaR and GRPO. The analysis quantifies both forward generation and training costs using established FLOP estimation formulas. To estimate computation costs, we employ two key formulas:

$$\text{Forward Generation Cost:} \quad X = 2 \times N \times D_{\text{rollout}}$$
$$\text{Training Cost:} \quad Y = 6 \times N \times D_{\text{train}}$$

$N$ represents the number of model parameters, $D_{\text{rollouts}}$ is the total number of tokens generated during inference, and $D_{\text{train}}$ is the total number of tokens used during training.

For the STaR baseline, we sampled 200 full rollouts per problem and selected solutions that correctly solved each problem. In contrast, for MRT (STaR), we generated just 1 complete rollout per problem, then selected 10 prefixes from this rollout and sampled 20 continuations for each prefix to approximate the information gain.

When applied to the NuminaMATH dataset containing 20,000 problems, using Llama-3.1-8B-Instruct with 4,000 token completions, this approach yielded 12,000 correct solutions for training (with incorrect solutions discarded). Training proceeded for three epochs. The total FLOP calculations are:

$$
\begin{aligned}
\text{STaR:} \quad & 2 \times 8\,\text{B} \times 200 \times 20\,\text{K} \times 4\,\text{K} + 6 \times 8\,\text{B} \times 12\,\text{K} \times 4\,\text{K} \times 3 \\
= \;& 256 \times 10^{18} + 6912 \times 10^{15} \\
= \;& 2.62912 \times 10^{20}\,\text{FLOPs} \\
\text{MRT (STaR):} \quad & 2 \times 8\,\text{B} \times (1 + 20 \times 10) \times 20\,\text{K} \times 4\,\text{K} + 6 \times 8\,\text{B} \times 12\,\text{K} \times 4\,\text{K} \times 3 \\
= \;& 25728 \times 10^{16} + 6912 \times 10^{15} \\
= \;& 2.64192 \times 10^{20}\,\text{FLOPs}
\end{aligned}
$$

For the GRPO baseline and MRT (RL), we used a different sampling strategy. In MRT (RL), we first generated 1 complete rollout per problem, then selected a prefix and sampled 10 rollouts to approximate information gain. In both methods, given a prompt, we sampled 4 responses and maximized their group advantage estimations. The resulting FLOP calculations are:

$$\begin{aligned}
\text{GRPO:} \quad & 2 \times 3\,\text{B} \times 20\,\text{K} \times 4 \times 4\,\text{K} + 6 \times 8\,\text{B} \times 20\,\text{K} \times 4 \times 4\,\text{K} \times 4 \\
= \quad & 192 \times 10^{16} + 6144 \times 10^{16} \\
= \quad & 6.336 \times 10^{19}\,\text{FLOPs} \\
\text{MRT (RL):} \quad & 2 \times 3\,\text{B} \times 20\,\text{K} \times (1 + 10 + 4) \times 4\,\text{K} + 6 \times 8\,\text{B} \times 20\,\text{K} \times 4 \times 4\,\text{K} \times 4 \\
= \quad & 720 \times 10^{16} + 6144 \times 10^{16} \\
= \quad & 6.864 \times 10^{19}\,\text{FLOPs}
\end{aligned}$$

Our analysis reveals that MRT (STaR) requires only $1.01\times$ more FLOPs $(2.64192 \times 10^{20}/2.62912 \times 10^{20})$ than STaR to achieve comparable performance, while using $1.7\times$ fewer tokens during inference. Similarly, MRT (RL) uses just $1.08\times$ more FLOPs than GRPO to achieve equivalent performance, while requiring 1.6× fewer tokens during inference.

These results demonstrate that our MRT approach achieves a favorable trade-off between computational cost and token efficiency, making it particularly valuable for deployment scenarios where inference efficiency is critical.

## F. Discussion and Future Work

While *MRT* shows strong empirical results, it raises a number of important questions that merit further study:

- **Choice of $\mu$ in *MRT*.** We choose $\mu$ as a greedy guesser based on the trace so far. Are there better meta-provers or reward parameterizations that can improve performance?

- **Characteristics of the base model.** All base models used here exhibit limited strategy diversity. Would models with broader reasoning strategies further amplify the benefits of regret minimization?

- **Branched rollout implementation.** Our reward is computed at the end of traces. Could more efficient implementations of branched rollouts reduce variance and improve learning?

- **Train-time vs test-time compute tradeoff.** While *MRT* uses more train-time compute, we hypothesize that it provides better test-time efficiency. We provide an initial "back-of-the-envelope" cost analysis in Appendix E. However, a formal, FLOPs-matched evaluation in the full online RL implementation of our approach remains an important direction.

**Follow-up work in the community on progress-based dense rewards.** Since our paper, more recent work in the community such as (Wang et al., 2025; An et al., 2025; Qi et al., 2025; Tu et al., 2025; Guo et al., 2025) also highlights the importance of utilizing dense rewards. In particular, the approach of focuses on identifying "critical" tokens (the authors call it "forking" tokens) based on per-token entropy of the next token distribution, but these are precisely tokens where advantages will be non-zero. Within our formulation, forking tokens will likely correspond to tokens that appear at the beginning of an episode that makes non-trivial progress towards or away from the solution the solution (and hence attains non-zero advantages). It appears that selectively modifying rewards on these tokens results in much larger performance improvements than 0/1 outcome-reward RL, further strengthening the point that dense reward signals of some form, applied to important tokens can be beneficial. Not only is the performance higher, but the gap between performance of this dense reward inspired approach and 0/1 outcome-reward RL increases as model size grows, further corroborating the promise of such approaches asymptotically. We believe that *MRT* should enjoy similar properties.

# G. Implementation Details

## G.1. Pseudocode

---

**Algorithm 1** *MRT* (STaR)

---

1: **Input** base model $\pi_{\theta_b}$; problems $\mathcal{D}$; reward function $r$
2: model $\pi_\theta \leftarrow \pi_{\theta_b}$, fine-tuning dataset $\mathcal{D}_{\text{ft}} \leftarrow \emptyset$
3: **for** iteration = 1, ..., T **do**
4:      **for** $x \in \mathcal{D}$ **do**
5:          Sample one rollout $z_{0:j} \sim \pi_\theta(\cdot|x)$
6:          Compute rewards $\{r^\mu_{\text{prg},i}\}^j_{i=1}$ for each prefix $z_{0:i}$ using Definition 4.1 for progress.
7:          **if** $\{r^\mu_{\text{prg},i}\}^j_{i=1} > 0$ **then**
8:              $i \leftarrow \arg\max^j_{i=0}\{r^\mu_{\text{prg},i}\}$
9:              Sample $y \sim \pi_\theta(\cdot|x, z_{0:i})$ s.t. $r(x,y) = 1$
10:             $\mathcal{D}_{\text{ft}} \leftarrow \mathcal{D}_{\text{ft}} \cup \{(x, z_{0:i}, y)\}$
11:          **end if**
12:      **end for**
13:      $\pi_\theta \leftarrow$ Fine-tune $\pi_\theta$ with $\mathcal{D}_{\text{ft}}$ and a negative log likelihood loss
14: **end for**

---

---

**Algorithm 2** *MRT* (RL)

---

1: **Input** base model $\pi_{\theta_b}$; problems $\mathcal{D}$; initialize model $\pi_\theta \leftarrow \pi_{\theta_b}$
2: **for** iteration = 1, ..., T **do**
3:      $\pi_{\text{ref}} \leftarrow \pi_\theta$
4:      **for** step = 1, ..., k **do**
5:          Sample a batch $\mathcal{D}_b$ from $\mathcal{D}$
6:          **for** $q \in \mathcal{D}_b$ **do**
7:              Sample one partial rollout $z_{0:j} \sim \pi_{\text{ref}}(\cdot|q)$, where $j$ is selected randomly
8:              Sample G rollouts $\{z^i_{j+1:}, y^i\}^G_{i=1} \sim \pi_\theta(\cdot|q, z_{0:j})$
9:              Compute rewards $\{r_i + \alpha \cdot r^\mu_{\text{prg},i}\}^G_{i=1}$ for each sampled output $(z^i_{j+1:}, y^i)$ using Definition 4.1 for progress and 0/1 correctness reward. The progress reward is computed using an additional set of $G$ rollouts that force the model to terminate.
10:          **end for**
11:          Update the policy $\pi_\theta$ via GRPO (Shao et al., 2024) with $\{r_i + \alpha \cdot r^\mu_{\text{prg},i}\}$ in place of $\hat{A}_i$
12:      **end for**
13: **end for**

---

## G.2. Hyperparameters for Open-ended Parameterizations

For *MRT* (STaR), we utilize the TRL codebase, but we customize the loss function to be weighted by progress defined in Definition 4.1. The base models are directly loaded from Hugging Face: DeepSeek-R1-Distill-Qwen-7B.

| Hyperparameter | Values |
| --- | --- |
| learning_rate | 1.0e-6 |
| num_train_epochs | 3 |
| batch_size | 256 |
| gradient_checkpointing | True |
| max_seq_length | 16384 |
| bf16 | True |
| num_gpus | 8 |
| learning rate | 1e-6 |
| warmup ratio | 0.1 |

*Table 2.* Hyperparameters used for *MRT* (STaR)

For *MRT* (RL), we utilize the open-r1 codebase, but we customize the loss function to be weighted by progress defined in Definition 4.1. The base models are directly loaded from Hugging Face: DeepSeek-R1-Distill-Qwen-1.5B and DeepScaleR-1.5B-Preview.

| Hyperparameter | Values |
| --- | --- |
| learning_rate | 1.0e-6 |
| lr_scheduler_type | cosine |
| warmup_ratio | 0.1 |
| weight_decay | 0.01 |
| num_train_epochs | 1 |
| batch_size | 256 |
| max_prompt_length | 4096 |
| max_completion_length | 24576 |
| num_generations | 4 |
| use_vllm | True |
| vllm_gpu_memory_utilization | 0.8 |
| temperature | 0.9 |
| bf16 | True |
| num_gpus | 8 |
| deepspeed_multinode_launcher | standard |
| zero3_init_flag | true |
| zero_stage | 3 |

*Table 3.* Hyperparameters used for *MRT* (RL)

### G.3. Hyperparameters for Backtracking Search

For *MRT* (STaR), we utilize the trl codebase, but we customize the loss function to be weighted by information gain defined in Definition 4.1. The base models are directly loaded from Hugging Face: Llama-3.1-8B-Instruct.

| Hyperparameter | Values |
|---|---|
| learning_rate | 1.0e-6 |
| num_train_epochs | 3 |
| batch_size | 256 |
| gradient_checkpointing | True |
| max_seq_length | 4096 |
| bf16 | True |
| num_gpus | 8 |
| learning rate | 1e-6 |
| warmup ratio | 0.1 |

*Table 4.* Hyperparameters used for *MRT* (STaR)

For *MRT* (RL), we utilize the open-r1 codebase, but we customize the loss function to be weighted by information gain defined in Definition 4.1. The base models are directly loaded from Hugging Face: Llama-3.2-3B-Instruct.

| Hyperparameter | Values |
|---|---|
| learning_rate | 1.0e-6 |
| lr_scheduler_type | cosine |
| warmup_ratio | 0.1 |
| weight_decay | 0.01 |
| num_train_epochs | 1 |
| batch_size | 256 |
| max_prompt_length | 1500 |
| max_completion_length | 1024 |
| num_generations | 4 |
| use_vllm | True |
| vllm_gpu_memory_utilization | 0.8 |
| temperature | 0.9 |
| bf16 | True |
| num_gpus | 8 |
| deepspeed_multinode_launcher | standard |
| zero3_init_flag | true |
| zero_stage | 3 |

*Table 5.* Hyperparameters used for *MRT* (RL)

# H. Additional Results

## H.1. More Results for Open-ended Parameterizations

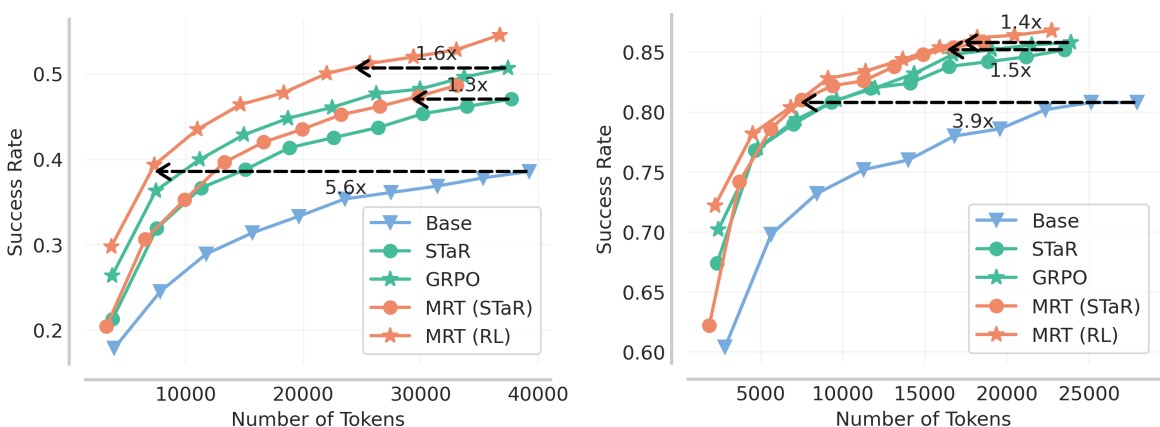

*Figure 17.* **MRT pass@k performance of R1-Distill-Qwen-1.5B with RL** on (Left) AIME; (Right) MATH500.

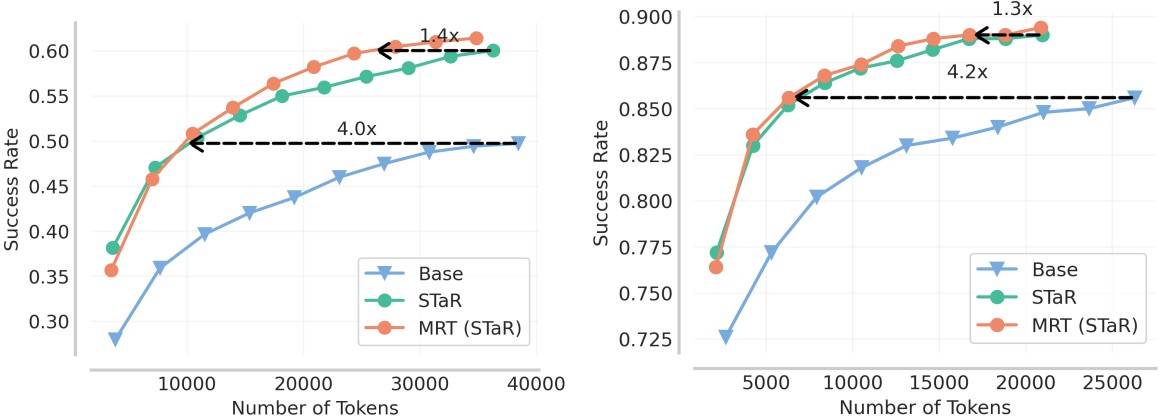

*Figure 18.* **MRT pass@k performance of R1-Distill-Qwen-7B with STaR**, on (Left) AIME; (Right) MATH500.

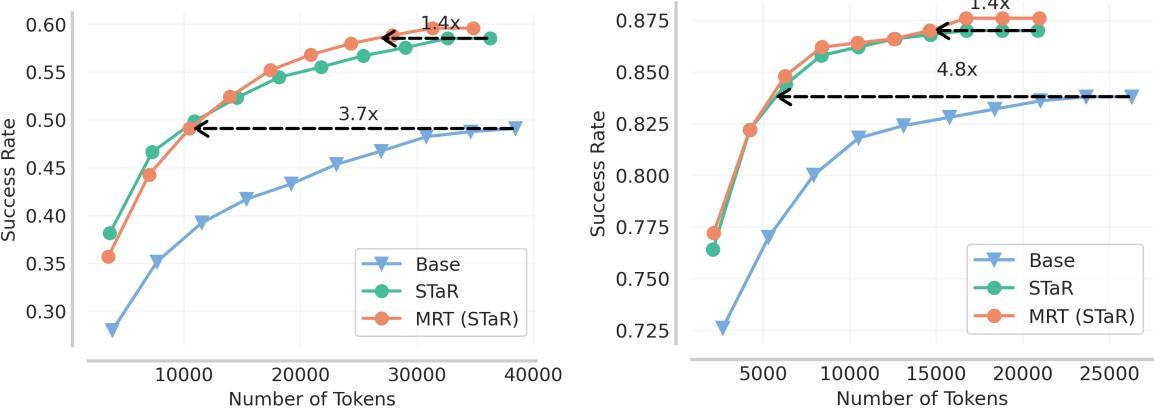

*Figure 19.* **MRT maj@k performance of R1-Distill-Qwen-7B with STaR** on (Left) AIME; (Right) MATH500.

## H.2. More Results for Backtracking Search

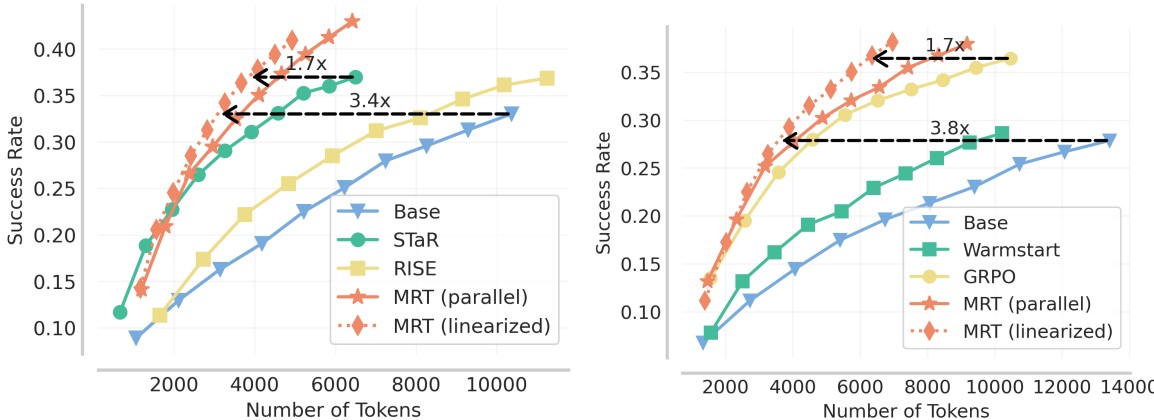

*Figure 20.* **MRT pass@k performance of R1-Distill-Qwen-1.5B** for k = 1, 2, ..., 10 on AIME (Left) STaR; (Right) RL. Observe that **MRT** attains the best performance as more tokens are sampled.

## I. Full Analysis of DeepSeek-R1

In this section we will give a more detailed outline on our analysis of DeepSeek-R1 derivates from Section I. We focus our analysis primarily on a subset of 40 problems taken from Omni-MATH. We chose Omni-MATH because it is not an explicit benchmark that DeepSeek-R1 reports (DeepSeek-AI et al., 2025) and is still challenging for many models. We chose 10 problems from each of the difficulty levels 4, 4.5, 5, and 5.5. The reason for doing this is to better capture the model's ability to make progress, which would not be apparent if the model got an accuracy near 0 or 100. We additionally also performed our analysis on the 30 problems from AIME 2024, which is a commonly-studied benchmark that we also report on in the main text.

The first step in our analysis is to generate solutions to problems with DeepSeek-R1-Distill-Qwen-32B, the model in the R1 family that we analyze. For each problem, we sample 4 responses at a temperature of 0.7 and 8192 maximum token length. We obtain our direct pass@k baseline with the same settings on Qwen2.5-32B-Instruct, except that we obtain 32 responses to simulate pass@32. Qwen2.5-32B-Instruct shares the same base model as DeepSeek-R1-Distill-Qwen-32B, but it is fine-tuned only on direct reasoning chains that do not employ thinking strategies such as backtracking and verification.

**Construction of episodes.** After we have obtained these initial completions, we separate them into episodes by filtering for explicit phrases that indicate a disruption in the natural flow of logic. We further constrain each episode to be at least three steps (each "step" is an entry separated by the delimiter "\n\n") to avoid consecutive trivial episodes. The explicit phrases are listed in Figure 21. If a step begins with one of these phrases, then we consider it to be the beginning of a new episode. The number of episodes depends on the problem and particular solution that was sampled. The distribution is shown in Figure 23. Due to the large number of episodes, we group the episodes into groups of 5 for Omni-MATH and groups of 3 for AIME, so each point on the blue curve in Figures 24 and 25 represents 5 or 3 episodes.

**Experimental setup.** For each prefix of episodes $\mathbf{z}_{0:j-1}$, where $j$ is a multiple of 5 or 3 respectively (as discussed in the previous paragraph), we ask the model to terminate its thinking, summarize its existing work, and give an answer. This is the way we approximate the computation of the best-guess policy $\mu(\cdot|\mathbf{x}, \mathbf{z}_{0:j-1})$, as discussed in Section I. To ensure a natural termination, we append the prompt shown in Figure 22 to the end of the prefix so that the model computes $\mu(\cdot|\mathbf{x}, \mathbf{z}_{0:j-1})$. This is repeated 8 times on every prefix to simulate maj@8, at temperature 0.7 and 4096 max tokens. Finally, we compute blue ($[\mathbf{maj@1}]_j$ at $j$ values) and green curves (for each $j$, $[\mathbf{maj@p}]_j$ at $p = 1, 2, 4, 8$) in Figures 24 and 25.

---

**Explicit step prefixes for separating episodes in R1 solution**

Wait
But wait
Alternatively
Is there another way to think about this?
But let me double-check
But hold on

---

*Figure 21.* **Explicit step prefixes for separating episodes in R1 solution**. This is a list of phrases that indicate a disturbance in the natural flow of logic under R1. If a step begins with one of these phrases, we consider it the start of a new episode.

---

**Prompt used to extract answer from R1**

{Insert $\mathbf{x}, \mathbf{z}_{0:j-1}$ here ($\langle$think$\rangle$ tag will be part of $\mathbf{z}_{0:j-1}$)}

Time is up.

Given the time I've spent and the approaches I've tried, I should stop thinking and formulate a final answer based on what I already have.
$\langle$\think$\rangle$

**Step-by-Step Explanation and Answer:**

1. **

---

*Figure 22.* **Prompt used to extract answer from R1**. We use the prompt above to simulate $\mu(\cdot|\mathbf{x}, \mathbf{z}_{0:j-1})$ and extract an answer after j episodes.

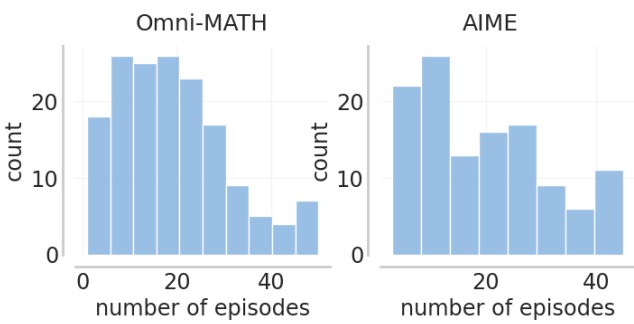

*Figure 23.* **Distribution of the number of episodes generated by R1 responses on AIME and Omni-MATH**.

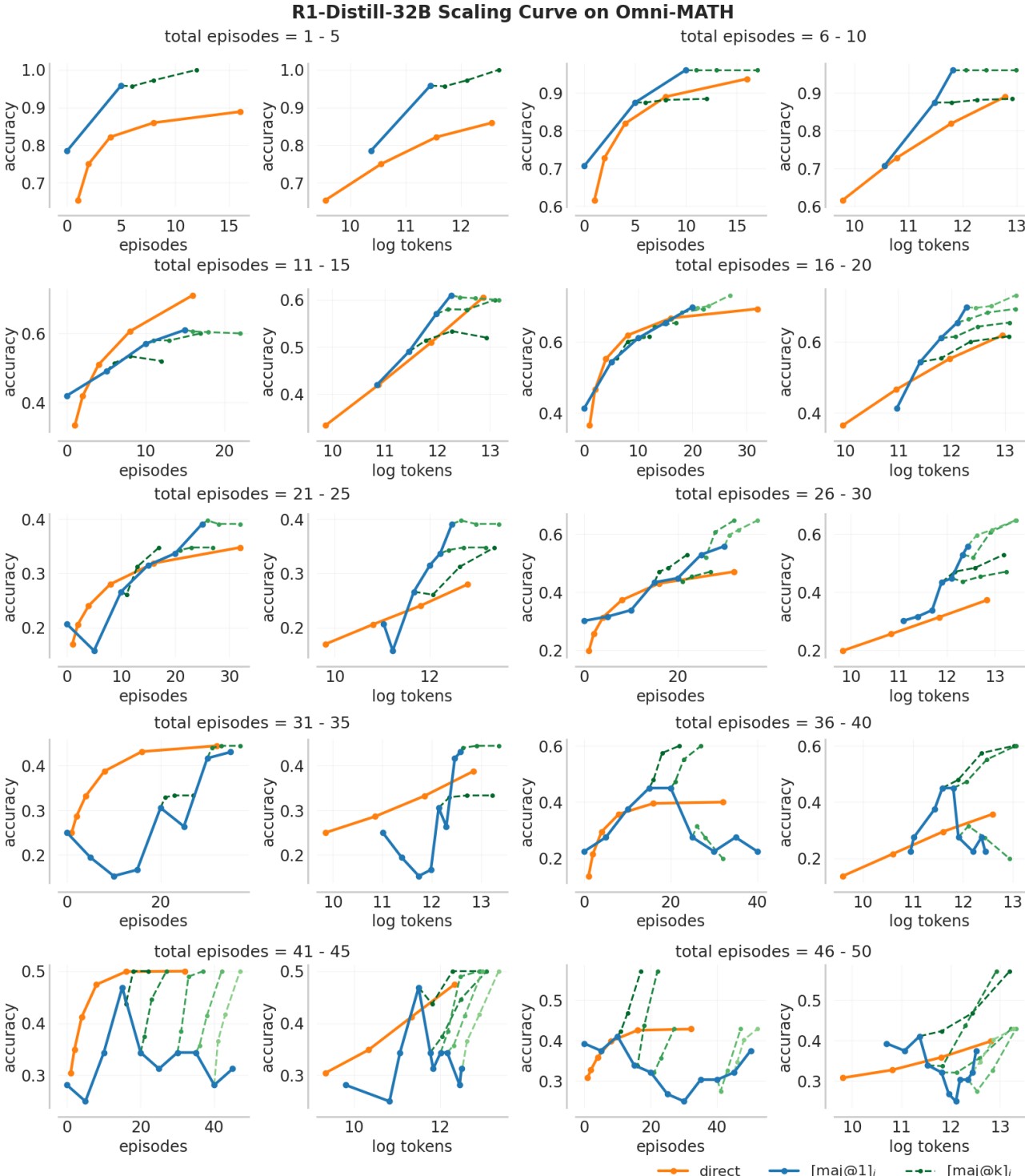

*Figure 24.* **DeepSeek-R1-Distill-Qwen-32B scaling curve on Omni-MATH subset across different episodes**. We compare scaling up the test-time compute for the R1-32B distilled model with **direct** pass@k for k = 1, 2, 8, 16, 32 against $[\mathbf{maj@p}]_j$ for p = 1, 2, 4, 8 and varying levels of $j$. Note that the total episodes matches the length of the blue curve. It is a range rather than a single number due to the concatenation of episodes into groups of 5 as mentioned in the full analysis.

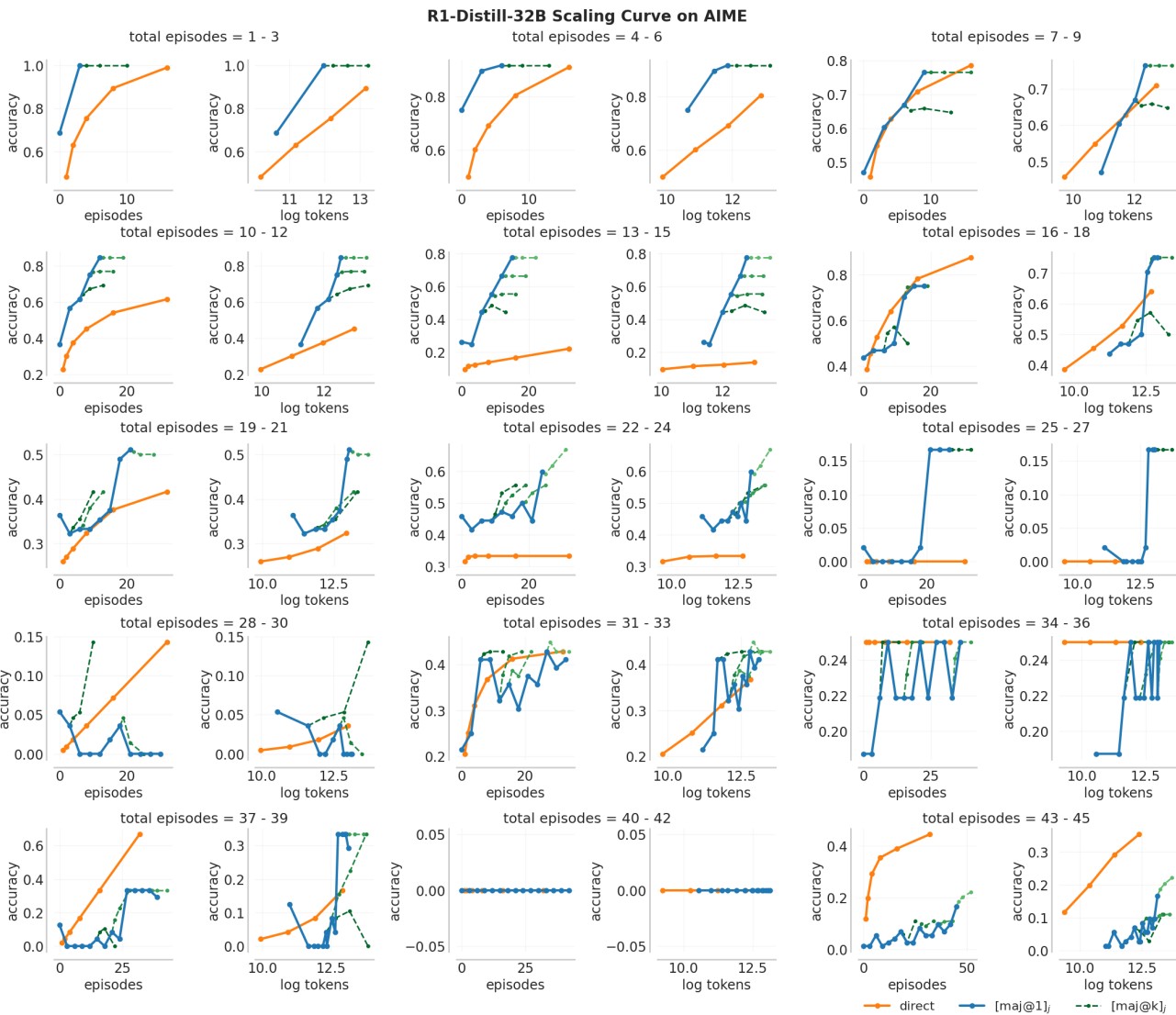

*Figure 25.* **DeepSeek-R1-Distill-Qwen-32B scaling curve on AIME 2024 across different episodes**. We compare scaling up R1 compute with **direct** pass@k for k = 1, 2, 8, 16, 32 against $[\mathbf{maj@p}]_j$ for p = 1, 2, 4, 8 and varying levels of $j$. It is a range rather than a single number due to the concatenation of episodes into groups of 3 as mentioned in the full analysis.

## J. Additional regret analysis of MRT models

In this section, we perform the analysis in the previous section on our own *MRT* STaR model fine-tuned from DeepSeek-R1-Distill-Qwen-7B to get a sense of its ability to make steady progress (Figure 27) and contrast it against the baseline of tuning DeepSeek-R1-Distill-Qwen-7B with STaR (Figure 28) (we repeat the same analysis in the RL setting but omit the intermediate figures since we already show the final results in Figure 26). We further condense these figures and extend the normalized regret analysis in Section 6.4.1 to answer the following question: *On different LLMs, how well does $[\textbf{maj@1}]_j$ (blue curves in Figure 27) with more episodes $j$ perform compared to $[\textbf{maj@k}]_{j'}$ (green curves in Figure 27) with fewer episodes $j'$?* In other words, do LLMs make meaningful progress through more sequential episodes compared to the alternative of stopping at an earlier episode and running maj@k?

To answer this, we augment the setting in our original regret analysis. Instead of using the theoretically optimal policy that achieves perfect accuracy in one episode, we take the optimal policy to be the best of maj@k from an earlier episode (green curve) and maj@1 from a later episode (blue curve). With this optimal policy, the regret is nonzero whenever a green curve lies above the blue curve, and zero otherwise (since, in regret, we subtract the optimal policy by the blue curve). The resulting regret measures the difference in performance between $[\textbf{maj@k}]_{j'}$ with fewer episodes $j'$ and $[\textbf{maj@1}]_j$ with more episodes $j$. Additionally, to get a sense of how each reasoning episode contributes to progress, we choose to look at the compute budget in episodes rather than tokens.

In both STaR and RL settings, we see that *MRT* gives the lowest normalized regret compared to the other approaches, implying more progress made in sequential episodes compared to maj@k on fewer episodes.

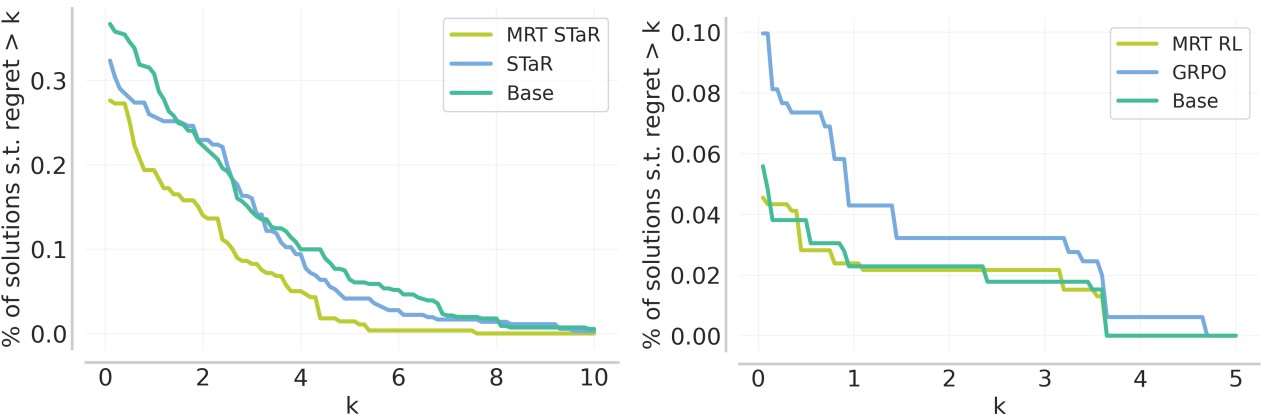

*Figure 26. **Normalized regret of different algorithms at different episode budgets**. **Left:** **MRT** (STaR) on DeepSeek-R1-Distill-Qwen-7B has a lower curve than STaR and Base models, indicating better progress in more sequential episodes compared to maj@k on fewer episodes. **Right:** **MRT** (RL) on DeepScaleR-1.5B-Preview also shows a lower curve compared to Base and GRPO, again demonstrating better progress in more sequential episodes.*

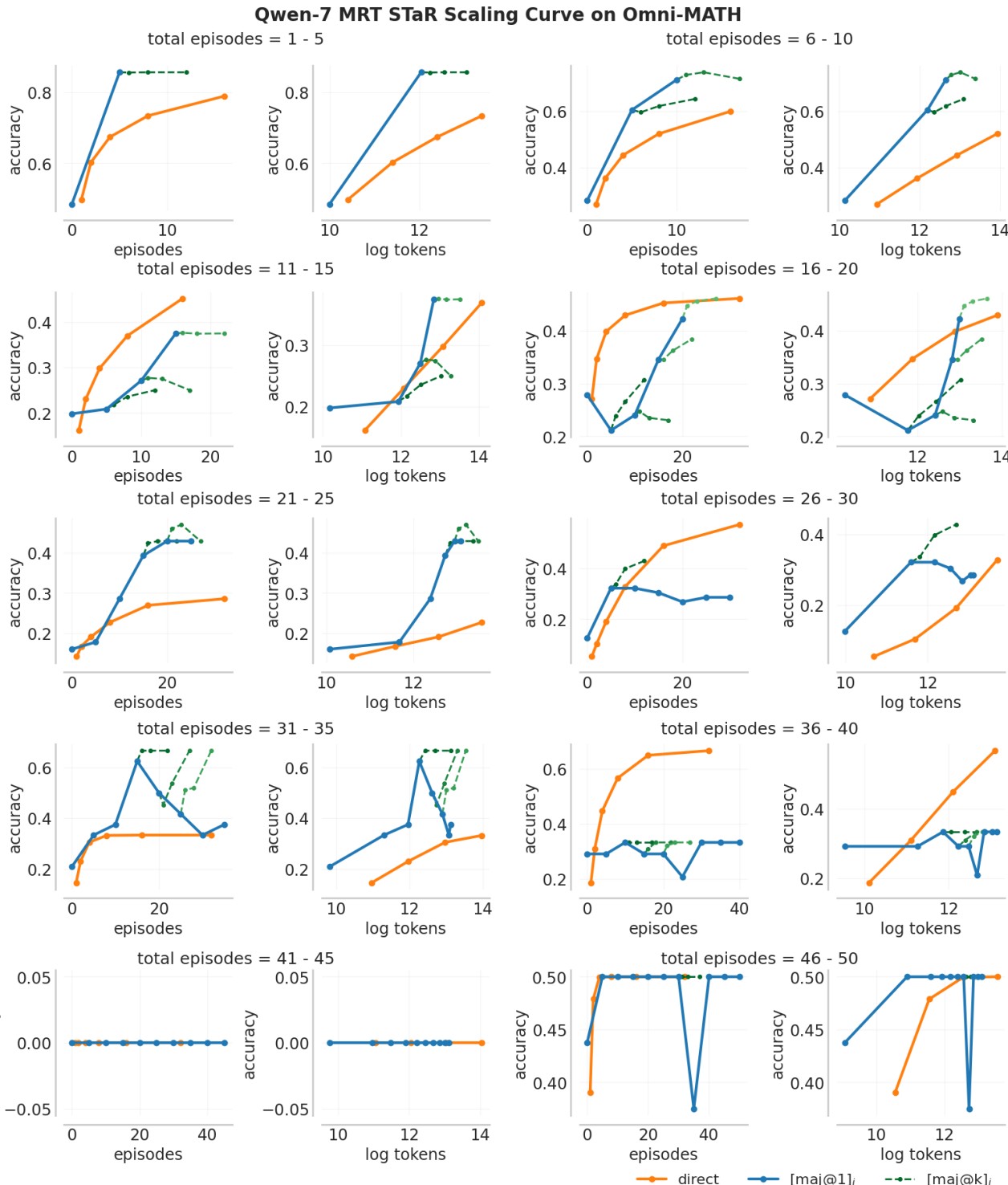

*Figure 27.* **MRT STaR (on DeepSeek-R1-Distill-Qwen-7B) scaling curve on Omni-MATH subset across different episodes**. We compare scaling up compute with the **direct** base model Qwen2.5-Math-7B-Instruct (orange curve) pass@k for k = 1, 2, 8, 16, 32 against $[\mathbf{maj@p}]_j$ for p = 1, 2, 4, 8 and varying levels of $j$ (blue curve and green curves).

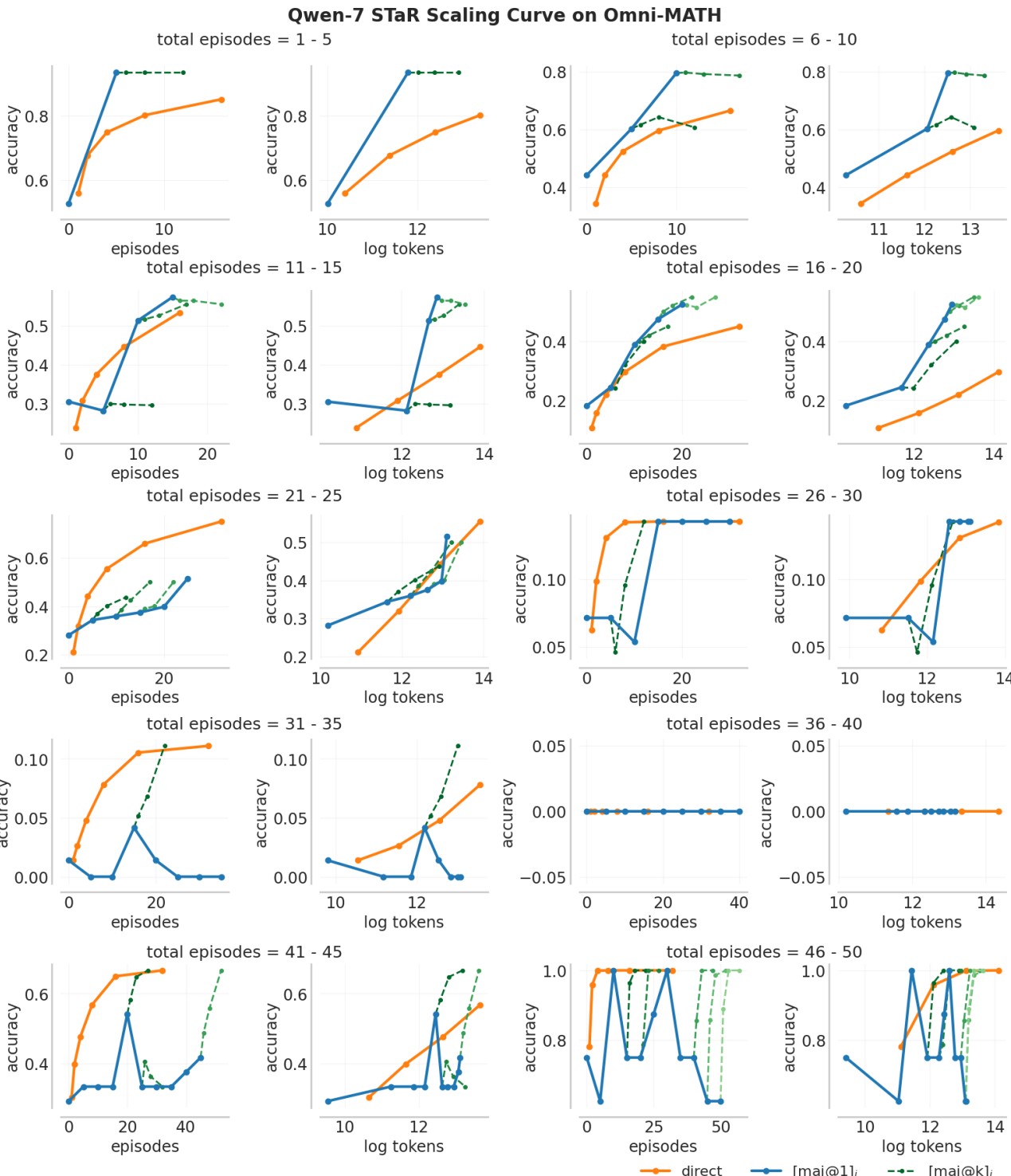

*Figure 28.* **STaR (on DeepSeek-R1-Distill-Qwen-7B) scaling curve on Omni-MATH subset across different episodes**. We compare scaling up compute with the **direct** base model Qwen2.5-Math-7B-Instruct (orange curve) pass@k for k = 1, 2, 8, 16, 32 against $[\mathbf{maj@p}]_j$ for p = 1, 2, 4, 8 and varying levels of $j$ (blue curve and green curves).

## K. Extrapolation Analysis

In this section, we extrapolate our model's test-time compute by using the budget-forcing technique from Muennighoff et al. (2025). This requires appending the token "Wait" to the end of the thought block to push the model to think more. For a given thought block, we experiment with doing this procedure 0/2/4/6/8 times, each time stopping when the closing $\langle \backslash \text{think} \rangle$ tag is produced or when we reach a maximum budget of 2048 tokens. To ensure that the model does not run into the scenario of endless repeating a phrase, we iterate through the options "Wait", "Alternatively", "But hold on", "But wait" as the "Wait" phrase to append to the end of the thought block. The results for the extrapolation on the Qwen-7B ***MRT*** (STaR) model and for the DeepScaleR-1.5B ***MRT*** (RL) model as shown in Figure 29. Note that the numbers do not exactly match the numbers in Table 1 due to randomness.

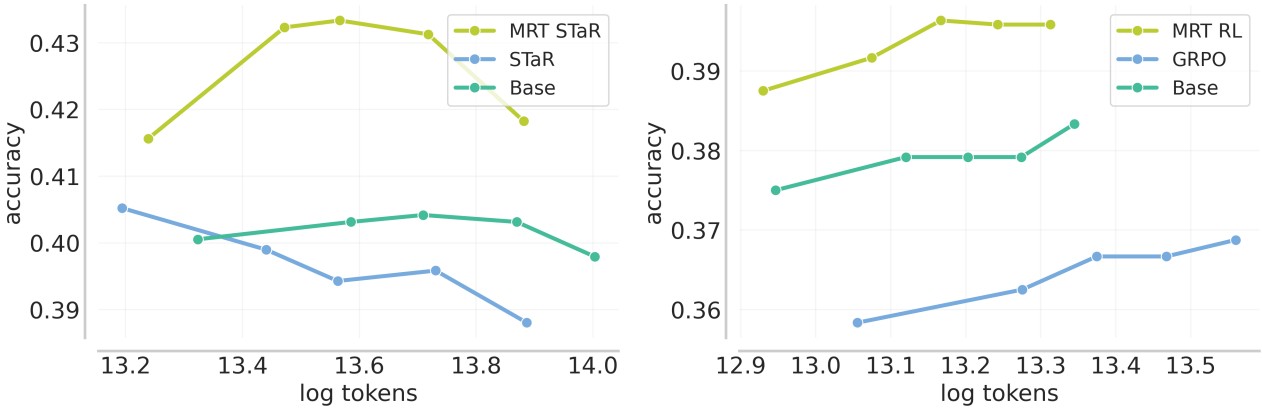

*Figure 29.* ***Extrapolation by pushing the model to think more with "Wait"***. **Left:** ***MRT*** (STaR). ***MRT*** (STaR) on DeepSeek-R1-Distill-Qwen-7B extrapolates better than the other two approaches when budget forcing 2/4/6 times, but the performance dips at 8 times, that said the performance of STaR decreases throughout. **Right:** ***MRT*** (RL) on DeepScaleR-1.5B-Preview without any extrapolation begins at a higher accuracy, but all approaches extrapolate similarly.

# L. Some Concrete Examples

## L.1. Backtracking Search

---

**Example of Backtrack**

**Query**: A type of operation is defined as follows: $\left(\begin{bmatrix} a & b \\ c & d \end{bmatrix} = ad - bc\right)$. The conjugate of the complex number $\left(\begin{bmatrix} 1+i & -1 \\ 2 & 3i \end{bmatrix}\right)$ is $\_\_\_\_\_\_$.

**Solution**: $\left(\begin{bmatrix} a & b \\ c & d \end{bmatrix}\right)$ is defined as $(ad - bc)$, but this doesn't appear to be a matrix operation. Instead, it seems to be a determinant operation. The problem statement seems to be a bit unclear, but if we consider the given complex number $\left(\begin{bmatrix} 1+i & -1 \\ 2 & 3i \end{bmatrix}\right)$ as a $2 \times 2$ matrix, we can calculate its determinant, which would be the conjugate.

Step 1: To find the conjugate of the complex number $\left(\begin{bmatrix} 1+i & -1 \\ 2 & 3i \end{bmatrix}\right)$, we first need to calculate the determinant of this matrix using the formula: $(ad - bc)$.

Step 2: From the given matrix, we have $a = 1 + i$, $b = -1$, $c = 2$, and $d = 3i$.

Step 3: The determinant of the matrix is $((1+i)(3i) - (-1)(2)) = (3i^2 + 2)$.

Step 4: Simplify the expression by substituting $(i^2)$ with $-1$: $(3(-1) + 2) = (-3 + 2)$.

Step 5: Calculate the expression: $(-3 + 2 = -1)$.

Step 6: The conjugate of a complex number $(a + bi)$ is $(a - bi)$, so the conjugate of $(-1)$ is $(-1)$.

Answer: $\boxed{-1}$ ✗

Wait, this seems off. Let's try something else.

Step 3: Plugging these values into the determinant formula, we get $(1+i)(3i) - (-1)(2) = 3i^2 + 3i + 2 = -3 + 3i + 2 = -1 + 3i$.

Step 4: However, the conjugate of a complex number is obtained by changing the sign of the imaginary part. Therefore, the conjugate of $-1 + 3i$ is $-1 - 3i$. Step 5: Therefore, the conjugate of the given complex number $\left(\begin{bmatrix} 1+i & -1 \\ 2 & 3i \end{bmatrix}\right)$ is $(-1 - 3i))$.

Answer: $\boxed{-1 - 3i}$ ✓

---

*Figure 30.* **Example of backtrack trajectory used to train the model.** The trajectory shows that the model first try to solve the problem, then it recognized that the prior solution is wrong from step 3, therefore, the model backtrack to step 2 in the prior solution and redo step 3 with correction. The mistake is highlighted in red, the correction is highlighted in green, and the backtracking step detection is highlighted in yellow.

## L.2. Open-Ended Parameterizations

---

**Example of Open-Ended Parameterizations**

**Query**: Point $(x, y)$ is randomly picked from the rectangular region with vertices at $(0, 0), (2008, 0), (2008, 2009)$, and $(0, 2009)$. What is the probability that $x > 2y$? Express your answer as a common fraction.

**Solution**:

¡think¿

Okay, so I have this problem where a point $(x, y)$ is randomly picked from a rectangular region with vertices at $(0, 0)$, $(2008, 0)$, $(2008, 2009)$, and $(0, 2009)$. I need to find the probability that $x > 2y$.

Hmm, maybe I should think of it as the area above the line $x = 2y$ within the rectangle. But actually, when $x > 2y$, for each $y$, $x$ starts from $2y$ up to 2008. So, the region is a trapezoid or a triangle?

Let me think. The line $x = 2y$ goes from $(0, 0)$ to $(2008, 1004)$. So, above this line, within the rectangle, is a region that is a triangle with vertices at $(0, 0)$, $(2008, 1004)$, and $(2008, 2009)$. Is that right?

Wait, no. If I consider $x > 2y$, then for each $y$, $x$ starts at $2y$ and goes to 2008. So, the region is actually a trapezoid with vertices at $(0, 0)$, $(2008, 1004)$, $(2008, 2009)$, and $(0, 2009)$. Hmm, but $(0, 2009)$ is not on the line $x = 2y$, so maybe it's a trapezoid.

Alternatively, when integrating over the region, you can think of slicing horizontally. For each y in $[0, 1004]$, $x$ runs from $2y$ to 2008. For $y$ in $[1004, 2009]$, there are no x satisfying $x >= 2y$ because $2y >= 2008$.

Therefore, the area where $x > 2y$ is the integral from $y = 0$ to $y = 1004$ of $(2008 - 2y)dy$

Time is up. Given the time I've spent and the approaches I've tried, I should stop thinking and formulate a final answer based on what I already have.

¡/think¿

First, total area is $2008 * 2009$.

Favorable area is the integral from 0 to 1004 of $(2008 - 2y)dy$.

Calculating the integral:

The integral of $2008dy$ is $2008y$.

The integral of $-2ydy$ is $-y^2$.

Therefore, evaluated from 0 to 1004:

$[2008 * (1004) - (1004)^2] - [0 - 0] = 2008 * 1004 - (1004)^2 = (2008 - 1004)1004 = 1004 \cdot 1004$.

Hold on, that's clever, it factors into 1004 squared.

So the favorable area is $1004^2$.

Therefore, the probability is $(1004^2)/(2008 * 2009)$.

The probability is $\boxed{502/2009}$ ✓

---

*Figure 31.* **Example of trajectory generated in the open-ended setting.** The trajectory shows how the model initially tries to conceptualize the problem within the "think" section. It changes its logical approach several times, and ultimately is forced to stop thinking and generate a solution.

