# OpenReview forum: "Optimizing Test-Time Compute via Meta Reinforcement Finetuning"
_ICML.cc/2025/Conference — ICML 2025 poster_

### Official Review · Reviewer_zHoJ · 2025-03-12

**Overall Recommendation:** 4

**Summary:**

The paper formalizes the problem of optimizing test-time compute as a meta-reinforcement learning problem and proposes to use cumulative regret as an optimizing objective instead of barely the outcome reward. The cumulative regret can be calculated by estimating the information gain. The authors further develop the meta-reinforcement finetuning (MRT) and show that MRT leads to substantial improvement on the AIME dataset.


## update after rebuttal

The authors have addressed my concerns and I will keep my score of 4 and suggest for acceptance.

**Claims And Evidence:**

The claims are generally supported by the experimental evidence in the paper.

**Essential References Not Discussed:**

It's probably good to at least mention OpenAI O1 model and cite their blog post.

**Experimental Designs Or Analyses:**

I checked all experimental designs and analyses. They look good to me. For additional comments, please refer to Other Strengths And Weaknesses.

**Methods And Evaluation Criteria:**

The proposed methods and evaluation criteria make sense.

**Other Comments Or Suggestions:**

Typos: repeated learning rate in table 1.

**Other Strengths And Weaknesses:**

The paper is generally well-written with interesting idea of formulating the test-time computing problem as a meta-RL problem. The idea and the proposed approach are both interesting and novel. The results are also solid. However, I find some of the writings confusing and can be further improved. Please see below for additional weaknesses.

1.	In Fig. 2 of the analyses of Deepseek-R1, the authors mix pass@k, majority@p and the episodes together, which makes this plot rather confusing at first glance. Also, I do not know how the authors make the plot for accuracy versus log-tokens. I would assume that different problems would have very different number of tokens per episode/pass. Those details are not clear even after reading the relevant appendices.

2.	The papers lack some details regarding how they perform the experiments. I suggest the authors adding the experimental details to the appendix. For example, the authors suggest that one can either use another LLM or the same LLM as the policy model to estimate the information gain. However, which LLM is used and how it is used to estimate the information gain is not clear from the paper for the experiments conducted.

3.	It is a little confusing for RL to run for multiple iterations since the authors use an online-RL framework. I guess the authors mean running with additional randomly sampled subset of the dataset. I suggest the authors clarifying this confusion.

**Questions For Authors:**

1.	In Fig. 6 and Fig. 7, the curves of MRT-B end earlier in number of tokens. Would it be helpful to steer the model to continue the episodes? Would it continuously improve the performance as reported in the s1 paper or would it saturate instead?

**Relation To Broader Scientific Literature:**

How to optimize the model to efficiently use additional test-time compute is a very important problem since the release of OpenAI-o1 model. The authors analyze the open-sourced reasoning model from a novel meta-RL perspective and propose new method that show significant improvement, which contributes to the development of more advanced reasoning model.

**Theoretical Claims:**

N. A.

---

> ### Author Rebuttal · Authors · 2025-04-01
>
> Thank you for your positive review of the paper! We will add more experimental details to the Appendix – in particular, regarding the experimental setting for both MRT (STaR), MRT (RL), and analysis on R1, and we will also cite the [o1](https://openai.com/o1/) and [o3](https://openai.com/index/openai-o3-mini/) blog post in the paper. We will address your concerns below, and would appreciate it if you would be willing to raise your score if you find your concerns addressed. We are happy to answer any remaining questions.
>
> > In Fig. 2 of the analyses of Deepseek-R1, the authors mix pass@k, majority@p and the episodes together, which makes this plot rather confusing at first glance. Also, I do not know how the authors make the plot for accuracy versus log-tokens. I would assume that different problems would have very different number of tokens per episode/pass. Those details are not clear even after reading the relevant appendices.
>
> It's correct that different problems would have very different numbers of tokens per episode and different numbers of episodes. Therefore, we first split the problems into different groups based on the number of episodes. For all solutions with a certain number of episodes (e.g., 6-10, 26-30), we average the number of tokens per episode and accuracy across those solutions. In other words, to plot the blue line, we fix j, find the average number of tokens up to j episodes, and then average maj@k given j episodes in the thought block across different solutions. Please let us know if this is clear, and we will add this discussion to the paper as well.
>
> > The authors suggest that one can either use another LLM or the same LLM as the policy model to estimate the information gain. However, which LLM is used and how it is used to estimate the information gain is not clear from the paper for the experiments conducted.
>
> Thanks for the question! We will use the one extra page in the final version to also include more details in the main paper and definitely add the rest to the appendix.
>
> In regards to your question above, we note that there are multiple ways to estimate the information gain or rewards. In this paper, we use Monte Carlo rollouts with the base model as the estimator. Specifically, to compute the reward of a given prefix, we sample multiple rollouts to complete this prefix with the base model. We then use the success rate among these rollouts to represent the reward of the prefix. The reason we mention that "one can either use another LLM or the same LLM as the policy model to estimate the information gain" is because Monte Carlo rollouts are not the only method to estimate rewards. One can also train a progress reward model to assess how effective a prefix is for a given problem, or use an LLM as a judge to provide the estimation.
>
> > It is a little confusing for RL to run for multiple iterations since the authors use an online-RL framework. I guess the authors mean running with additional randomly sampled subset of the dataset. I suggest the authors clarifying this confusion.
>
> Yes, and multiple iterations are also useful to update the reference model to prevent it from being too different / off-policy.
>
> > Would it be helpful to steer the model to continue the episodes with MRT? Would it continuously improve the performance as reported in the s1 paper or would it saturate instead?
>
> **New results in more open-ended settings:**
>
> To extend MRT to more episodes, we ran MRT directly on top of distilled variants of DeepSeek-R1. We refer to this setting as the "open-ended" setting, since the episodes now – much like our analysis in Section 4 – are not constrained to follow a specific format. We defer the training details to the [supplement](https://sites.google.com/view/icml25mrt/home#h.hx11wcth8wc3). We evaluated MRT and GRPO on AIME 2024/25, and AMC 2023 datasets (20 samples per problem) from different base models in the [supplement](https://sites.google.com/view/icml25mrt/home#h.d85qg8xhjc12). Our models fine-tuned on DeepScaleR-1.5B-Preview achieve state-of-the-art performance for their size: **47.2% success on AIME 2024 and 39.7% on AIME 2025**. Across multiple base models, MRT's relative performance improvement is about **2–3x** compared to outcome-reward RL (GRPO).
>
> We also measure the cumulative regret metric for both MRT and other baselines (STaR, GRPO, base model). To do so, we first run the R1 analysis on our own models and take the optimal policy $\pi^*$ to be the one that achieves perfect accuracy in one episode. Intuitively, we are computing the red area (denoting cumulative regret) normalized at different cutoff points/token budgets. As shown in the [supplement](https://sites.google.com/view/icml25mrt/home#h.nllm3x1c80pa), models trained with MRT have the lowest regret. Moreover, in extrapolation regions where we steer our trained model to think more with "Wait" (similar to S1), the performance of **MRT doesn't plateau but continues to improve**.

---

> > ### Comment · Reviewer_zHoJ · 2025-04-03
> >
> > I thank the authors for their responses to my questions. Including additional experimental details in the paper is important for clarity and understanding. The quality of the paper would be significantly improved if these details were incorporated. I have no further concerns and recommend that the paper be accepted.

---

> > > ### Author Response · Authors · 2025-04-03
> > >
> > > Thank you for your thoughtful feedback. We're glad that we've addressed all your concerns and appreciate your recommendation for acceptance! We will certainly incorporate all the additional experimental details into the final version of the paper as suggested.
> > >
> > > Since our revisions have addressed your concerns and will improve the paper's quality, we respectfully ask if you might consider raising your evaluation score to reflect these improvements. And thank you again for your valuable input throughout this review process.

---

### Official Review · Reviewer_GafA · 2025-03-13

**Overall Recommendation:** 4

**Summary:**

This paper introduces Meta Reinforcement Finetuning (MRT), a framework to optimize how large language models (LLMs) utilize test-time computational resources.  The authors frame test-time compute optimization as a meta reinforcement learning (RL) problem, where the LLM generates a stream of token episodes (e.g., reasoning steps, backtracking attempts) to solve a query. The goal is to minimize cumulative regret, a metric measuring how effectively intermediate episodes contribute to discovering the correct answer. MRT augments standard outcome-based RL (e.g., 0/1 correctness) with a dense reward bonus based on the information gain from each episode. This reward quantifies the utility of intermediate steps in reducing uncertainty about the final answer. The results demonstrate that optimizing for cumulative regret via MRT enables LLMs to balance exploration and exploitation, improving both efficiency and generalization to larger test-time budgets. The framework is scalable and adaptable to diverse reasoning strategies beyond backtracking.

**Claims And Evidence:**

**MRT-B Improves Efficiency and Performance**

Combing MRT-B with STaR and GRPO, experiments on AIME show 30% and 38% token efficiency gains over baselines (Figures 7–8). However, in Figure 10, the information gain bonus underperforms length-penalized RL in reducing the completion length. Actually, the length does not decrease compared with GRPO at all on the right side of Figure 10. Why the length is not decreased? Didn't MRT increase token efficiency?

Besides, although the paper's analyze seems suitable for multiple episodes, but in the pseudocode in Appendix B, the experiments are only on z_0, z_{1:2}, only two episodes? It might be a drawback of this paper if I understand this correctly.

**Existing Methods (e.g., DeepSeek-R1) Exhibit High Regret**

Section 4 demonstrates that DeepSeek-R1’s accuracy plateaus or degrades with more episodes Figures 2. The figure is a little bit hard to understand, I will say my understanding and you can say if it is correct. The 6 dots on the "direct" line are pass@k=1,2,4,8,16,32. You break the reasoning at step 0,5,10,15,20... to compute maj@p, p=1,2,8. But I see three green dots in each green line, so I assume it should be p=1,2,4,8? There is no green line breaching from early episode since at early episode p>1 is the same as p=1. So your result is basically saying that the in later episodes, it is better to do parallel sampling than sequential sampling.

Besides, I checked Appendix C, The Omni-MATH subset (40 problems) and AIME (30 problems) are small and may not reflect broader generalization.

**Essential References Not Discussed:**

The references are properly discussed.

**Experimental Designs Or Analyses:**

See previous reviews.

**Methods And Evaluation Criteria:**

**Information Gain Calculation**

In definition 5.1,  to calculate the information gain, it requires to sample repeatedly at the end of episodes, will that be the major computational cost in the training?

**Equation (2)**

The introduction of information gain into the rl is interesting. But how is $c_k$ generated? what is the relationship between $z_1, z_2\cdots, z_{k-1}$, they are independently generated by pi conditioned on $c_k$? And how is $k$ defined here?

**Warmstart SFT’s**

In Sec 6.2, this paper proposed to construct a warmstart dataset. I think the desription of the construction is not clear. Using Figure 5 as an example, using what metric (in math equation) you choose node 2, from node 2, can we construct something like 0-2-6-13-5-9-14-5-9-15, i.e., two wrong answer before one correct ones. Besides, why your construction is easy to fit, is it because of common prefix of correct and wrong answer? If the prefix was shared more, such as 0-2-6-11-17-16, will this be easier to fit?

**Other Comments Or Suggestions:**

See previous reviews.

**Other Strengths And Weaknesses:**

See previous reviews.

**Questions For Authors:**

This paper proposes an interesting method in test-time-compute, but some writings of the methods can be improved. I am willing to raise my score if my concerns are resolved.

**Relation To Broader Scientific Literature:**

N.A.

**Theoretical Claims:**

There are no theoretical claims.

---

> ### Author Rebuttal · Authors · 2025-04-01
>
> Thank you for your feedback! To address your concerns, we clarify the notion of token efficiency and interpret the results of MRT in comparison with length-penalty and baseline GRPO, add new results running MRT on top of DeepSeek-R1 models to extend beyond three-episode setting, and add numerous visualizations of regret with and without MRT to justify the efficacy of the regret metric. **Please let us know if your concerns are addressed, and if so, we would be grateful if you could raise your score.**
>
> > [New expt.] Results for multiple episodes
>
> To extend MRT to more episodes, we ran MRT directly on top of distilled variants of DeepSeek-R1. This “open-ended setting” removes the constraints on the episodes to follow a specific format, making them freeform similar to our analysis in Section 4. We defer the training details to the [supplement](https://sites.google.com/view/icml25mrt/home#h.hx11wcth8wc3).
>
> **Results:** We evaluated MRT and GRPO on AIME 2024/25, and AMC 2023 datasets from different base models in [supplement](https://sites.google.com/view/icml25mrt/home#h.d85qg8xhjc12). Our models fine-tuned on DeepScaleR-1.5B-Preview achieve state-of-the-art performance for their size: **47.2% success on AIME 2024 and 39.7% on AIME 2025**. Across multiple base models, MRT's relative performance improvement is about **2–3x** compared to outcome-reward RL (GRPO).
>
> **Comparisons to length penalty:** We also run an additional comparison on top of the DeepScaleR-1.5B model, where we apply an explicit length penalty but fine-tune it with GRPO. In agreement with findings in the submission, we find that incorporating a length penalty results in worse pass@1 accuracy.
>
> In the [supplement](https://sites.google.com/view/icml25mrt/home#h.39szyztelvn4), we also measure the cumulative regret of MRT, GRPO/STaR, and base models. To do so, we choose $\pi^*$ to be the one that achieves perfect accuracy within one episode. Intuitively, we are computing the red area in the [supplement](https://sites.google.com/view/icml25mrt/home#h.ytlrapcrku0y) normalized for different token budgets. MRT attains smallest regret, even when extrapolating beyond training budget (similar to [s1](https://arxiv.org/abs/2501.19393)).
>
> > Token efficiency and length penalty
>
> To clarify, **token efficiency refers to maximum performance at minimal tokens.** The tradeoff of using length penalty is that although it reduces the number of tokens substantially, the performance plateaus beyond a point (e.g., see Figure 10, when tokens > 8000). MRT surpasses it when we allow a larger number of tokens, and the model's performance continues to increase. In addition, note that in the above experiments, MRT even outperforms length penalty in terms of pass@1 performance.
>
> > Computation cost of MRT
>
> In supplement, we compute the total FLOPs used by MRT and STaR/GRPO for sampling and training. For NuminaMATH (20,000 problems) with Llama-3.1-8B-Instruct, STaR required 2.62×10²⁰ FLOPs while MRT needed 2.64×10²⁰ FLOPs (1.01× more) while attaining 1.7× fewer inference tokens. Similarly, GRPO used 6.34×10¹⁹ FLOPs versus MRT's 6.86×10¹⁹ FLOPs (1.08× more) but MRT used 1.6× fewer inference tokens. MRT uses <8% more computation while requiring 60% fewer tokens during inference to achieve the same performance.
>
> > Understanding of Figure 4
>
> Yes, your understanding is correct. To add to your point, this result is saying that sequential sampling does not realize the full potential of tokens spent, as the naive strategy of parallel sampling (maj@k) outperforms sequential thinking. When in principle, sequential thinking should easily express maj@k with the same number of tokens.
>
> > How is $c_k$ generated?
>
> Sorry for the confusion, it should be $c_j$, which consists of the prefix (first j episodes) sampled from the previous checkpoint. $k$ is defined as the number of episodes of the rollout. To avoid confusion, we provide a more detailed explanation of update in the [supplement](https://sites.google.com/view/icml25mrt/home#h.3d34xjyebemx).
>
> > Sec 6.2, warmstart dataset construction and other construction schema.
>
> In Figure 5, for each node from 0-13, we compute the information gain by using Definition 5.1, and select the one that maximizes the information gain. And yes, we can construct in other formats as suggested. Motivated by this, we extend the method to an open-ended setting.
>
> > Besides, I checked Appendix C, The Omni-MATH subset (40 problems) and AIME (30 problems) are small and may not reflect broader generalization.
>
> As shown in the [supplement](https://sites.google.com/view/icml25mrt/home#h.sej9e58f36gr), we added more results by evaluating Deepseek-R1 on AIME problems from 2015-2024 (293 problems in total). Our findings from the submission still hold, where the performance of off-the-shelf models does not improve as the thinking budget increases, and simple early termination with parallel sampling outperforms, but the model couldn't discover such solution.

---

> > ### Comment · Reviewer_GafA · 2025-04-07
> >
> > Thanks for providing additional experiments. I have changed my score to 4.

---

> > > ### Author Response · Authors · 2025-04-07
> > >
> > > Thank you for your valuable feedback. We’re glad that we’ve addressed all your concerns and greatly appreciate your recommendation for acceptance. As suggested, we’ll ensure that all the additional experimental details are incorporated into the final manuscript.

---

### Official Review · Reviewer_TSJb · 2025-03-13

**Overall Recommendation:** 3

**Summary:**

This paper suggests a novel perspective on test time compute through long generation through the formulation of meta-rl. It suggests that the correct way to trade exploration and exploitation, in this case, is through the notion of cumulative regret. Furthermore, it claims that we should judge if a partial response (=an episode) contributed to the overall success of the trajectory through a notion of information gain. The information gain can be used as an additional reward term, which leads to the MRT algorithm - a variant of either STaR or GRPO with the additional reward bonus. The method is evaluated on mathematical reasoning tasks (NuminaMATH and AIME datasets) and shows improved accuracy and token efficiency compared to standard outcome-reward RL.

**Claims And Evidence:**

In Figure 8, it seems like the difference between GRPO (Iter 2) and MRT (Iter 2) is only 1-2%. Can you please provide numerical values with CI so we will be able to understand if the gains are actually statistically significant?

**Essential References Not Discussed:**

None.

**Experimental Designs Or Analyses:**

Figure 2 confuses me. First, when you calculate maj@p for DeepSeek-R1. Do you sample the episodes p times or just the final response provided by \mu p times? If the latter, why does maj@p appears in the plot, as it contains multiple episodes? I understand why it is plotted as requiring more tokens but not more episodes.
In addition, my understanding is that maj@p is just a way to get a better estimation of the regret by constructing a more accurate \mu. If this is indeed the idea, why compare accuracy? Why the y axis in Figure 2 is not regret?
In general, the paper pushes the use of cumulative regret as a metric but doesn’t include any plot or other quantitive results that use it as a metric. This is one of the things that bother me the most.

**Methods And Evaluation Criteria:**

Equation 2 is not written in a clear way - does c_k contain k or j episodes?
It also doesn’t align with the description of the  MRT-B (RL) algorithm. There, You use \pi_old as \mu, to get an estimation of the information gain of z_0. This is unlike equation 2, where \pi_old is used to sample context. Please clarify it.

**Other Comments Or Suggestions:**

In Figure 4, it says once that J_r(\mu(\cdot|z_1,z_0,x) equals 0.5 and the other time it equals 0.75, and the third time that it is 0.25. Is it a typo?

**Other Strengths And Weaknesses:**

I think the perspective of test-time computing as a meta-rl problem is novel and interesting to the community. In addition, the reward shaping term suggested in the paper is grounded and seems helpful to the training process.
In addition, the paper provides a significant amount of experiments, checking the reward term on top of two popular algorithms - STaR and GRPO.

There are two main reasons I currently tend to reject the paper: the fact that the proposed metric is not used in either the analysis of current algorithms or the evaluation of the new one. And the fact that I'm not sure how significant the gain from MRT is compared to the baselines.

**Questions For Authors:**

MRT-B requires running meta-prover rollouts to estimate information gain, adding significant computational overhead to the training. For large-scale tasks or bigger models, it could become expensive to run repeated queries at intermediate steps. The paper would benefit from a more thorough discussion of how costly this is.

Both versions of MRT were trained to improve only the first rollout (z_0). Do you think extending the algorithm to evaluate the information gain of z_2, z_4,… will result in even bigger gains? I understand that such runs can be computationally expensive, therefore I’m not necessary expect numerical results.

**Relation To Broader Scientific Literature:**

The paper mentions STaR multiple times without ever explaining the algorithm. I think spending a few lines explaining it (maybe in section 2?) will make the paper easier to read for people who are not well-versed in the literature.

**Theoretical Claims:**

- Line 261 claims that “[utilizing previous policy] allows us to improve over the previous policy provably” but doesn’t provide proof.
- You introduce a reward bonus term to the RL problem, as defined in equation 2. Will the optimal policy for the augmented reward is the same as for the original one? I think this is an important thing to clarify.

---

> ### Author Rebuttal · Authors · 2025-04-01
>
> Thank you for your feedback! We've added new results measuring regret for MRT and baselines, showing MRT attains smaller regret and improved performance in more general settings. We will also update the paper with more clarifications and the definition of STaR. **If your concerns are addressed, we'd be grateful if you could raise your score.**
>
> > Significance of maj@k results in Figure 2 and measuring regret
>
> **Motivation for measuring maj@k.** Our goal with maj@k is not to compute regret, but to demonstrate a simple baseline using partial thinking traces that outperforms sequential thinking with more episodes. If sequential thinking worked effectively, it should have outperformed this basic maj@k approach (blue >> green).
>
> **Regret measurements:** We initially didn't measure regret because, similar to RL, this requires comparison against an optimal policy π* that's unknown beforehand. Performance over more episodes served as our proxy. However, we can consider the optimal comparator π* to be the one that achieves perfect accuracy in one episode. Here, the regret is the area between the blue / green / orange lines in Figure 2 and the horizontal line at y = 1. For each point on the R1 scaling curve, we can plot corresponding regret, normalized by episodes or tokens used. As shown in the [supplement](https://sites.google.com/view/icml25mrt/home#h.nllm3x1c80pa), the regret of direct and [maj@k]_j are lower compared to [maj@1]_j on solutions with more episodes, indicating that sequential episodes do not use tokens efficiently compared to majority voting from an earlier episode or the direct model.
>
> > Performance gain over baselines from MRT
>
> We want to highlight that the metric is not just pass@1 performance, but also token efficiency. We redrew the plots in the [supplement](https://sites.google.com/view/icml25mrt/home#h.ex6mhnuz4w7y) by omitting iter1 and highlighting token efficiency. With linearized evaluation, MRT achieves the same performance as GRPO with 1.6x fewer tokens.
>
> **New results in more open-ended settings:** We extended MRT to distilled variants of DeepSeek-R1, where episodes aren't constrained to follow a specific format. We evaluated MRT and GRPO on AIME 2024/25, and AMC 2023 datasets (20 samples per problem) from different base models in [supplement](https://sites.google.com/view/icml25mrt/home#h.d85qg8xhjc12). Our models fine-tuned on DeepScaleR-1.5B-Preview achieve state-of-the-art performance for their size: **47.2% success on AIME 2024 and 39.7% on AIME 2025**. Across multiple base models, MRT's relative performance improvement is about **2–3x** compared to outcome-reward RL (GRPO).
>
> The 95% confidence interval for our method fine-tuned from DeepScaleR-1.5B-Preview on AIME2024 is **±0.13%.** Given the stable estimation from 20 samples, we prioritized evaluating on more problems and omit the CI for other models.
>
> > Computational overhead in MRT training.
>
> As in [supplement](https://sites.google.com/view/icml25mrt/home#h.i3vh26dgt29u), we compute the total FLOPs used by MRT and STaR/GRPO for sampling and training. For NuminaMATH (20,000 problems) with Llama-3.1-8B-Instruct, STaR required 2.62×10²⁰ FLOPs while MRT needed 2.64×10²⁰ FLOPs (1.01× more) while attaining 1.7× fewer inference tokens. Similarly, GRPO used 6.34×10¹⁹ FLOPs versus MRT's 6.86×10¹⁹ FLOPs (1.08× more) but MRT used 1.6× fewer inference tokens. MRT uses <8% more computation while achieving same performance with 60% fewer tokens during inference.
>
> > Equation 2 clarification
>
> Thanks for pointing this out. It should be $c_{j}$​, which consists of the prefix (first j episodes) sampled from the previous checkpoint $\pi_\text{old}$. We provide a more detailed explanation of updated equation 2 in the [supplement](https://sites.google.com/view/icml25mrt/home#h.3d34xjyebemx).
>
> > clarify $\pi_\text{old}$ and $\mu$
>
> Policy $\mu$ can be any LLM (e.g., an "-instruct" model which is told to utilize episodes so far to guess the best answer). For implementation simplicity we use the success rate of Monte-Carlo rollouts on $\pi_\text{old}$ to represent $\mu$. We will add this clarification to the paper.
>
> > proof of claim in Line 26
>
> The theoretical argument behind this line is from Section 3 of the [TRPO paper](https://arxiv.org/abs/1502.05477), which shows optimizing policy under the state distribution induced by the old policy with a KL-constraint on actions results in monotonic performance improvement.
>
> > optimality of augmented reward
>
> As shown in the [supplement](https://sites.google.com/view/icml25mrt/home#h.ytlrapcrku0y), the optimal policy for the augmented reward (Right, w/ information gain) will also attain maximal reward under the original reward (Left, w/o information gain). However, not every policy that achieves the highest original reward exhibits maximal information gain. To see this intuitively, note that Equation (1) only guarantees correctness of the outcome, whereas maximal information additionally does so quickly.

---

> > ### Comment · Reviewer_TSJb · 2025-04-03
> >
> > Thank you to the authors for their response. Their clarification helped me better understand the proposed method, and the new results support their claims.
> > However, the empirical gains from the proposed method still appear marginal compared to GRPO, especially considering the added complexity of the training procedure.
> > I will update my score accordingly.

---

> > > ### Author Response · Authors · 2025-04-08
> > >
> > > Thank you for your response and for increasing your score! We are glad that our clarifications helped in this case.
> > >
> > > Regarding the bit that “the gains from the proposed method appear marginal compared to GRPO,” we would like to highlight several important considerations in regards to our results.
> > >
> > > First, it’s worth noting that the base models in our new results were already trained with RL on potentially a larger superset of prompts, or distilled from RL-trained models. Given this initialization, we should expect the gains from any subsequent fine-tuning to be modest in absolute magnitude, due to (1) fine-tuning an already fine-tuned model which often is known to result in entropy collapse, and (2) we had to use this initialization due to a lack of compute for fine-tuning a base model from scratch. Despite this high starting point, MRT still demonstrates:
> > > - Statistically significant and systematic gains with MRT that are 2-3× larger than those achieved by GRPO
> > > - Approximately 1.7× improvement in token efficiency
> > > - Less than 8% computation overhead from outcome-reward training
> > >
> > > The primary aspect of MRT is to optimize dense rewards when using test-time compute. Concurrent work outside of optimizing test-time compute from the process reward model literature, [PAVs](https://arxiv.org/abs/2410.08146), also shows that dense rewards are more effective than outcome rewards, especially on hard problems (Figure 8b in https://arxiv.org/pdf/2410.08146) due to improved exploration. We hypothesize that this kind of a difference between using dense rewards in MRT vs outcome reward training via GRPO on hard problems will also carry over in our experiments, if given enough compute and training time, which we are in lack of due to limited computational power.
> > >
> > > **New proposed experiment:** While we cannot run this experiment with dense rewards in time for the response since our compute for doing so was not available, we plan to run an experiment by re-running training with the DeepScaleR-1.5B recipe from scratch on their prompt mixture using dense rewards prescribed by MRT for the final (in contrast to fine-tuning the DeepScaleR-1.5B checkpoint). In addition, we will add a didactic experiment showing the efficacy of MRT on countdown (Game of 24) for the final version.

---

### Decision · Program_Chairs · 2025-05-01

**Decision:**

Accept (poster)

**Comment:**

Summary: This paper frames the optimization of test-time computation in large language models as a meta-reinforcement learning problem, advocating for the use of cumulative regret as the optimization objective, which is estimated through information gain. The authors introduce Meta Reinforcement Finetuning (MRT) and demonstrate its effectiveness in improving performance on the AIME dataset.

Review summary:
Reviewers agree that the paper presents a novel framing of test-time compute as a meta-RL problem, and that the proposed information-gain based shaping is principled. However, two reviewers questioned the magnitude and statistical significance of the reported gains—particularly between MRT and GRPO.

Recommendation & justification
I recommend acceptance. The core idea regarding how to use test-time budget wisely is novel. The conceptual contribution, experimental  data and improved token efficiency makes this paper worth accepting.